# Functional connectivity subtypes associate robustly with ASD diagnosis

**Sebastian GW Urchs[1,2]\*, Angela Tam[2], Pierre Orban[3,4], Clara Moreau[2,5,6], Yassine Benhajali[2], Hien Duy Nguyen[7], Alan C Evans[1], Pierre Bellec[2]\***

[1]Montreal Neurological Institute and Hospital, McGill University, Montreal, Canada; [2]Centre de Recherche de l'Institut Universitaire de Gériatrie de Montréal, University of Montreal, Montreal, Canada; [3]Centre de Recherche de l'Institut Universitaire en Santé Mentale de Montréal, Montreal, Canada; [4]Département de Psychiatrie et d'Addictologie, Université de Montréal, Montreal, Canada; [5]Sainte Justine Research Center, University of Montreal, Montreal, Canada; [6]Human Genetics and Cognitive Functions, Institut Pasteur, UMR3571 CNRS, Université Paris Cité, Paris, France; [7]School of Mathematics and Physics, University of Queensland, St. Lucia, Australia

**Abstract** Our understanding of the changes in functional brain organization in autism is hampered by the extensive heterogeneity that characterizes this neurodevelopmental disorder. Data driven clustering offers a straightforward way to decompose autism heterogeneity into subtypes of connectivity and promises an unbiased framework to investigate behavioral symptoms and caus-ative genetic factors. Yet, the robustness and generalizability of functional connectivity subtypes is unknown. Here, we show that a simple hierarchical cluster analysis can robustly relate a given individual and brain network to a connectivity subtype, but that continuous assignments are more robust than discrete ones. We also found that functional connectivity subtypes are moderately associated with the clinical diagnosis of autism, and these associations generalize to independent replication data. We explored systematically 18 different brain networks as we expected them to associate with different behavioral profiles as well as different key regions. Contrary to this predic-tion, autism functional connectivity subtypes converged on a common topography across different networks, consistent with a compression of the primary gradient of functional brain organization, as previously reported in the literature. Our results support the use of data driven clustering as a reliable data dimensionality reduction technique, where any given dimension only associates moder-ately with clinical manifestations.

**\*For correspondence:**
sebastian.urchs@gmail.com (SGWU);
pierre.bellec@criugm.qc.ca (PB)

## Editor's evaluation

The authors examine autism subtypes using functional connectivity data derived from magnetic resonance imaging. Autism spectrum disorder is notoriously heterogeneous, so the clustering approach to decompose this heterogeneity is attractive, however, the robustness of this approach and the generalization of groupings is unknown. The authors find that functional connectivity subtypes correspond to clinical autism diagnostic groupings and generalize using independent repli-cation data. Functional connectivity patterns are robust, but the discrete assignment of individuals to a group is moderate and suggests that the findings may reflect compression of the primary gradient of functional brain organization.

## Introduction

Autism spectrum disorder (ASD) is a prevalent neurodevelopmental condition of impaired social communication and restrictive behaviour, diagnosed in about 1% of children (*Lai et al., 2014*; *Baio et al., 2018*), that is associated with extensive heterogeneity of behavioral symptoms and neurobiological endophenotypes (*Jacob et al., 2019*; *Lombardo et al., 2019*). Functional magnetic resonance imaging (fMRI) has emerged as a promising technology to identify potential biomarkers of functional connectivity (FC) in ASD and other psychiatric disorders (*Castellanos et al., 2013*). However, efforts to characterize the functional brain organization in ASD have so far largely focused on case-control comparisons, thus ignoring the presumed heterogeneity of FC alterations (*Nunes et al., 2019*; *Hahamy et al., 2015*).

Data driven cluster analysis has long been proposed as a solution to decompose the heterogeneity of behavioral symptoms in ASD into distinct subtypes (*Eaves et al., 1994*; *Beglinger and Smith, 2001*), but these subtypes have proven difficult to distinguish in clinical practice (*Lord et al., 2012*) and were recently abandoned in favour of the broader concept of an autism spectrum (American Psychiatric Association. and DSM-5 *American Psychiatric Association, 2013*). The lack of progress toward reproducible, brain-based biomarkers of ASD (*Lombardo et al., 2019*) has renewed interest in clustering methods to decompose the heterogeneity of brain alterations into distinct subtypes that are hypothesized to underlie the multitude of behavioral symptoms. Note that we explicitly refer to brain-based functional connectivity subtypes as FC subtypes throughout this manuscript, to help distinguish them from behavioral subtypes derived from clinical symptoms.

To date, only a small number of studies have applied brain-based clustering to characterize the neurobiological heterogeneity in ASD and relate it to behavioral symptoms (*Hong et al., 2019b*). Early work on subcortical volume alterations in ASD has distinguished four subtypes, but did not find significant differences of behavioral symptoms between them (*Hrdlicka et al., 2005*). A more recent multi-modal analysis distinguished three subtypes of structural brain alterations in ASD and found that core ASD symptoms could be much better predicted from the structural MRI data when separate prediction models were trained on each subtype compared to the full, unstratified dataset (*Hong et al., 2018*). Work on the heterogeneity of FC in individuals with ASD, attention deficit hyperactivity disorder (ADHD), and neurotypical controls (NTC) distinguished three FC subtypes among regions in the default mode network (DMN) and found that each FC subtype was associated with all three diagnostic groups, indicating that these FC subtypes may be shared across diagnostic boundaries (*Kernbach et al., 2018*). An analysis of whole-brain FC in ASD and NTC individuals distinguished two FC subtypes of diverging within- and between-network connectivity, but similarly showed that the assignment of individuals to these FC subtypes was not associated to their clinical diagnosis (*Easson et al., 2019*).

These initial findings of imaging subtypes in ASD leave several important questions open. Firstly, studies have so far interpreted imaging subtypes both as discrete categories (*Hrdlicka et al., 2005*; *Hong et al., 2018*) and a continuous, multifactorial spectrum (*Kernbach et al., 2018*; *Easson et al., 2019*). However, the stability of either of these two methods of assigning individuals to subtypes has not been systematically established. Secondly, several previous studies have limited their investigation of imaging subtypes to individuals who were already diagnosed with ASD (*Hrdlicka et al., 2005*; *Hong et al., 2018*; *Tang et al., 2020*). It has not been clearly established whether imaging subtypes associated with ASD symptoms are specifically found among these diagnosed individuals, or are also prevalent in the general population. Behavioral symptoms in ASD overlap with those of other neurodevelopmental disorders and also extend into the general population (*Constantino and Todd, 2003*; *Grzadzinski et al., 2011*). Similarly, neurobiological endophenotypes associated with ASD have been shown to exist among individuals with other neuropsychiatric disorders (*Park et al., 2018*; *Di Martino et al., 2013*; *Moreau et al., 2020*). It is therefore important to investigate whether imaging subtypes identified in mixed samples of both ASD and NTC individuals show an association with ASD diagnosis and symptoms. Thirdly, none of the imaging subtypes associated with ASD in the literature have been replicated to date. The recent failure to replicate promising reports of clinically meaningful neuroimaging subtypes in depression (*Drysdale et al., 2017*; *Dinga et al., 2019*) has highlighted the importance of this limitation for the autism literature.

In this work, we aim to address the three outlined gaps by applying a straightforward, unsupervised clustering approach to subdivide a heterogeneous sample of both ASD and NTC individuals into their

network based FC subtypes. Firstly, we systematically evaluate the robustness of the FC subtype maps, and the discrete and continuous assignment of individuals to them. Secondly, we determine whether diagnosis naive FC subtypes show an association with clinical ASD diagnosis at the network level. And thirdly, we determine the generalizability of our findings by replicating them on an independent dataset.

To identify FC subtypes, we selected hierarchical agglomerative clustering because it is a well established method of clustering which — although not the most recent — has been widely used in the literature. As our aim was to understand how the classical concept of clinical subtypes translates to FC subtypes, we relied on a very established method rather than a more recent and possibly more performant approach.

Alterations of FC in different brain networks have been previously linked to different behavioral and cognitive symptoms of ASD (*Rudie et al., 2012*; *Cerliani et al., 2015*; *Abbott et al., 2016*). We thus hypothesized that identifying FC subtypes separately for individual brain networks may reveal connectivity profiles that are associated with distinct behavioral symptom profiles. To probe the connectivity of functional brain networks, we defined FC seed regions with a functional connectivity derived brain parcellation (*Urchs et al., 2017*) that has been shown to perform well in ASD related classification tasks (*Dadi et al., 2019*).

## Results

### Almost all individuals were assigned to a data driven FC subtype

We identified FC subtypes for each of the 18 non-cerebellar seed networks of the MIST_20 parcellation (*Urchs et al., 2017*). FC subtypes were extracted by a data driven clustering algorithm. The number of FC subtypes was also data-driven, based on a threshold of the homogeneity of FC maps in a subtype and a minimum number of individuals within a FC subtype. As a result, some individuals may not be assigned to a FC subtype. Specifically, the individual seed-FC maps were first corrected for covariates of non-interest (recording site, age, head motion) and centered voxel-wise to the sample mean. We then computed the spatial dissimilarity between individual seed-FC maps as 1 minus their spatial correlation (i.e. two perfectly anti-correlated maps would have a dissimilarity of 2, and two perfectly correlated maps a dissimilarity of 0). FC subtypes were then identified by hierarchical clustering on the between-individual dissimilarity matrix according to two criteria: the average spatial dissimilarity within a FC subtype was smaller than 1 and at least 20 individuals were part of the FC subtype (see *Figure 1* for an overview of the workflow). Across all 18 seed networks, we identified 87 FC subtypes in the discovery dataset (see Functional connectivity subtypes associate robustly with ASD diagnosis for a topographic overview). In each seed network we identified between 3 (medial visual network) and 6 FC subtypes (lateral visual network) that satisfied the criteria (the median number of FC subtypes was 5). The average number of individuals in a FC subtype was $\bar{N} = 79.4, (13.2 SD)$ out of a total sample size of $\bar{N} = 388$. On average across networks, 97% of the individuals in the discovery dataset were assigned to a FC subtype (Functional connectivity subtypes associate robustly with ASD diagnosis). The largest number of individuals not assigned to any FC subtype was 19, found with the inferior temporal gyrus seed network, and all individuals were assigned to FC subtypes in the ventral somatomotor and perigenual anterior cingulate seed networks. We thus show that the majority of individuals in the discovery dataset contributed to the identified 87 FC subtypes.

### FC subtypes are not driven by confounds and are robust to hyper-parameter choices

We characterized FC subtypes by three measures:

- *Discrete assignments* of individuals to FC subtypes, computed through clustering,
- *FC subtype maps* computed as the average seed-FC map of individuals discretely assigned to this FC subtype,
- *Continuous assignments of individuals* to FC subtypes, computed as the spatial correlation between the FC subtype map and each individual seed-FC map.

To ensure that FC subtypes did not simply reflect confounds in the data, we conducted additional analyses for each FC subtype measure. We found that the distribution of data collection sites across

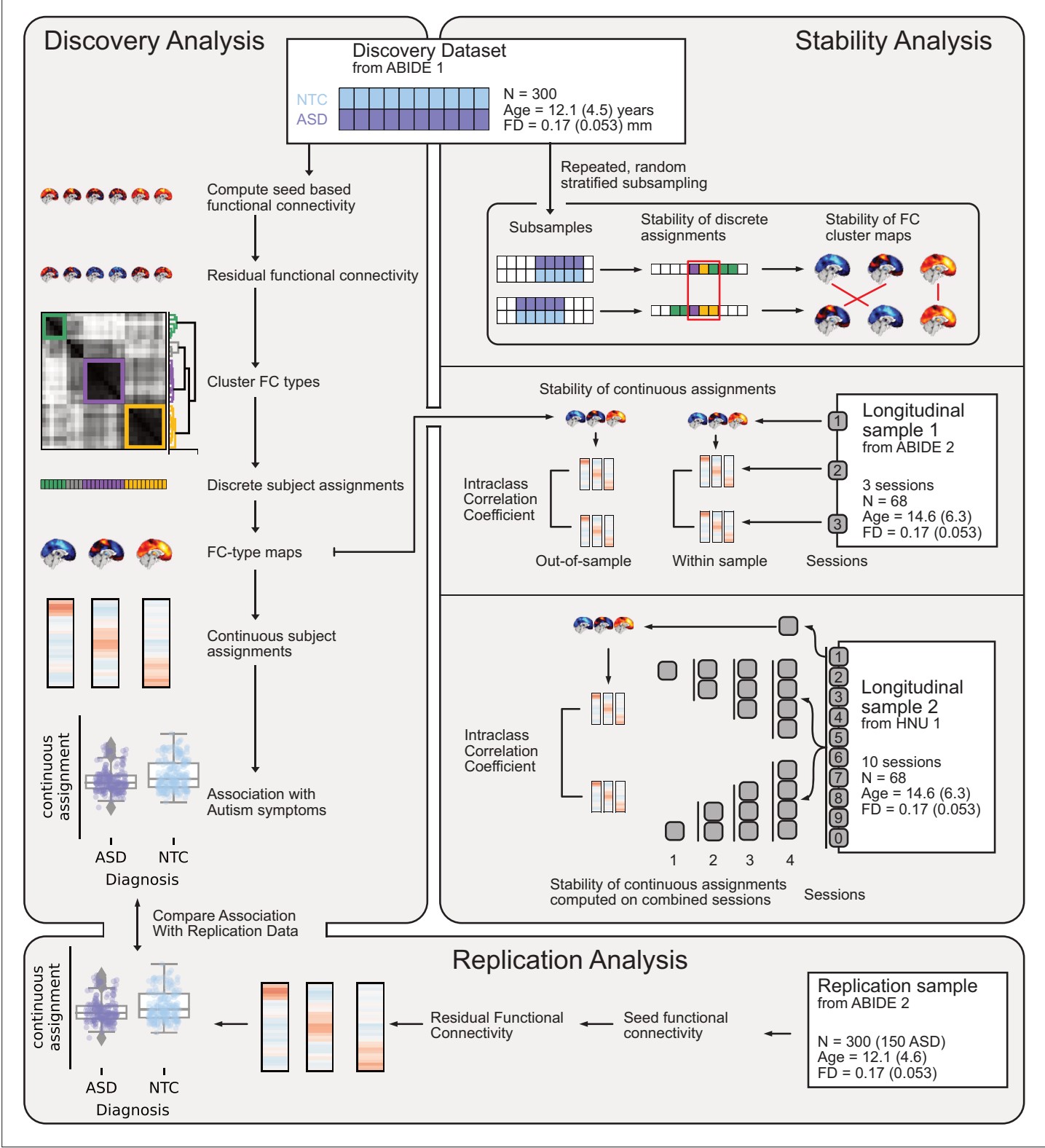

**Figure 1.** Overview of datasets and analyses presented in this work. Discovery Analysis: the discovery sample was drawn from ABIDE 1. Subtypes maps and continuous subtype assignments were extracted (left, middle) from the same data and associated with ASD diagnosis (left, bottom). Replication analysis: the replication sample was drawn from ABIDE 2. Continuous subtype assignments were extracted for subjects from the replication sample, using subtypes from the discovery sample. These continuous subtype assignments were again associated with ASD diagnosis (left, bottom) to replicate the discovery findings. Stability analyses: three stability analyses were conducted, using three different datasets. Stability of discrete subtype

*Figure 1 continued on next page*

**Figure 1 continued**
assignments and of subtype maps (right, top) was estimated using random subsamples of the discovery dataset to regenerate the subtyping process. Stability of continuous subtype assignments in an ASD sample was estimated across scan sessions from a longitudinal subsample of ABIDE 2 (right, middle). Continuous subtype assignments were either computed for subtypes extracted from a session of the same sample (within-sample) or from the discovery sample (out-of-sample). Finally, the impact of data availability on continuous subtype assignment stability was estimated across ten scan sessions of the longitudinal HNU1 dataset (right, bottom). Subtypes were extracted from one session of the dataset, and continuous subtype assignments were computed on individual or averaged sessions (2, 3, or 4).

discrete FC subtype assignments did not differ significantly from chance (Pearson's chi-square test), even when p-values were not adjusted to correct for the number of tests (the median uncorrected p-value was $p_{median} = 0.77$). FC subtype maps were robust across dissimilarity thresholds and higher thresholds led to the inclusion of smaller proportions of the sample (see *Appendix 2—figure 1* for an illustration). Lastly, we found no significant linear relationship between continuous assignments of individuals to FC subtypes and in-scanner head motion, and age for any seed network (Pearson's correlation). We thus show that FC subtype measures in the discovery dataset were not significantly driven by confounds.

## FC subtype maps are stable

We next aimed at evaluating the stability of FC subtype maps. For this purpose, we repeated the unsupervised clustering analysis used to generate FC subtypes on 1000 random subsamples of 50% of the discovery dataset. We then randomly drew 1000 pairs of subsamples and for each FC subtype map in one subsample, we found a best matching FC subtype map in the other subsample that had the highest spatial correlation (see Stability analysis). The average spatial Pearson correlation between matched FC subtype maps was $\bar{r} = 0.65$ (0.034$SD$) across all seed networks and subsamples. We observed small variations across seed networks: from $r = 0.58$ (0.081$SD$) for the inferior temporal gyrus seed network up to $r = 0.7$ (0.069$SD$) for the dorsal motor network (see *Figure 2*). We thus showed that the spatial maps of the identified FC subtypes were stable across perturbations in the discovery dataset.

## Discrete individual assignments to FC subtypes are not stable

We evaluated the stability of discrete assignments of individuals to a FC subtype. Specifically, for each individual, we measured the Dice similarity metric (*Dice, 1945*) between replicated FC subtypes including this individual across pairs of subsamples (see Stability analysis). The average overlap of discrete FC subtype assignments was low at $Dice = 0.22$ (0.025$SD$). That is, 22% of the FC subtype neighbours of an individual in one subsample would on average also be neighbours of this individual in another subsample. The range of overlap between FC subtypes was $Dice = 0.2$ (0.018$SD$) for the auditory network to $Dice = 0.28$ (0.046$SD$) for the medial visual network (see *Figure 2*). We thus showed that the discrete assignment of an individual to a FC subtype was not stable to perturbations in the discovery dataset.

**Table 1.** The 18 non-cerebellar MIST_20 seed network names and their abbreviation.

| Abbreviation | Network name |
| --- | --- |
| BG_THAL | Basal ganglia and thalamus |
| MOTnet_v | Somatomotor network ventral |
| ORBcor_NACC | Orbitofrontal cortex and nucleus accumbens |
| PGACcor_VMPFcor | Perigenual anterior cingulate cortex and ventromedial prefrontal cortex |
| ITgyr_Tpol | Inferior temporal gyrus and temporal pole |
| FPTCnet | Fronto parietal task control network |
| AUDnet_PINS | Auditory network and posterior insula |
| MVISnet | Medial visual network |
| AMY_HIPP_Pisul | Amygdala and hippocampus and peri insular sulcus |
| MOTnet_d | Somatomotor network dorsal |
| VATTnet_m | Ventral attention network medial |
| DMnet_ap | Default mode network anteriorposterior |
| DMnet_pm | Default mode network posteromedial |
| LVISnet | Lateral visual network |
| VVIS_DVIS | Ventral visual stream and dorsal visual stream |
| DMnet_l | Default mode network lateral |
| VATTnet_l | Ventral attention network lateral |
| FPnet | Fronto parietal network |

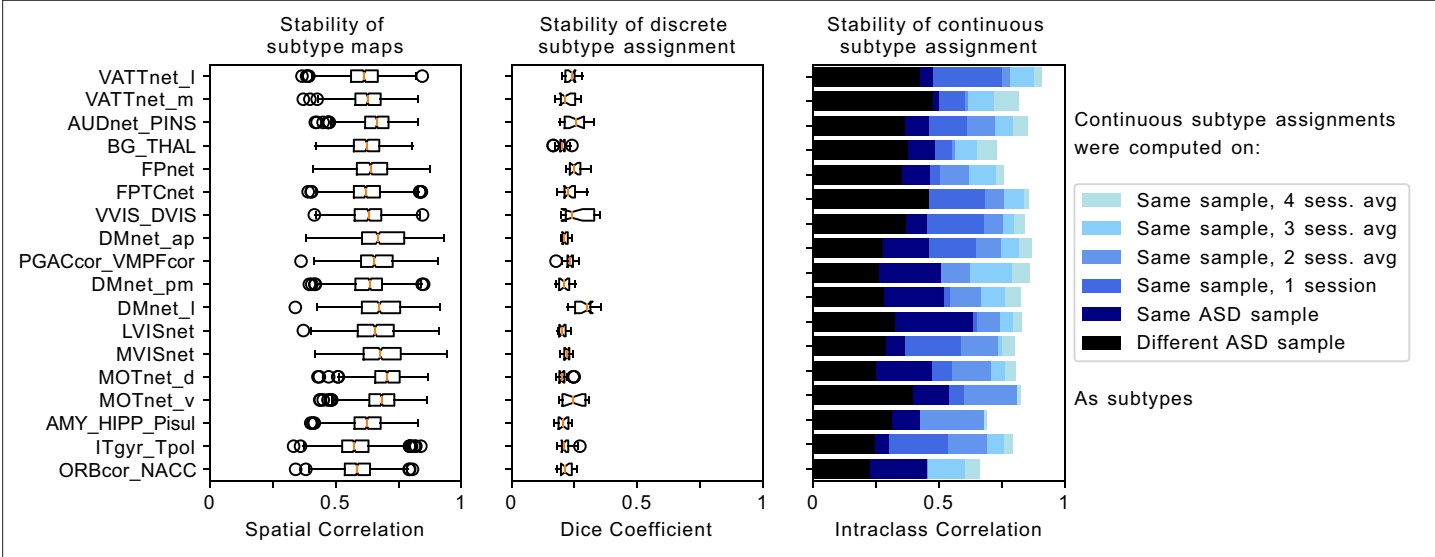

**Figure 2.** Robustness of FC-subtype outcomes across brain networks. Left column: Stability of the spatial FC-subtype maps. Boxplots represent the range of the average similarity between FC-subtype maps of the same brain network that were extracted from 1000 separate subsamples of the discovery dataset (each subsample included half of the sample n=194). MIST_20 seed network names are abbreviated, see *Table 1* for full seed network names. Middle column: Stability of discrete assignments to FC-subtypes. Boxplots represent the average overlap between the clusters an individual was assigned to in two different random subsamples. Right column: Stability of continuous assignments of individuals to a FC-subtype across repeated imaging sessions. Bar plots represent the average Intraclass Correlation between continuous assignments to FC-subtypes computed on separate longitudinal imaging sessions. The color hue reflects the data that continuous assignments were computed on: (black, out-of-sample) completely separate ASD datasets from the one used to compute subtypes, (dark blue, within-sample) same dataset but different scan session from the one used to compute subtypes, (blue to light blue) within-sample FC-subtypes in a general population data set where multiple scan sessions were combined to compute continuous FC-subtype assignments.

## Continuous individual assignments to FC subtypes are stable

We evaluated the stability of continuous assignments of individuals to FC subtypes for each seed network. To do so, we computed the intraclass correlation coefficient (ICC, *Shrout and Fleiss, 1979*) of the continuous assignments across repeated scan sessions. The observed ICC coefficients were interpreted (*Cicchetti, 1994*) as

- poor if less than 0.4
- up to 0.59
- up to 0.74
- if larger than 0.75

We first computed the ICC for within-sample stability, that is when both the individuals and FC subtypes maps were taken from the same data. We used data from Longitudinal sample 1 (a separate set of individuals in the ABIDE 2 sample for whom 3 scan sessions were available). We then identified FC subtypes on one scan session, computed the continuous assignments of individuals to these FC subtypes on the remaining two sessions, and computed the ICC of these continuous assignments across the two sessions (see *Figure 1* for an overview of the analysis datasets, and Stability analysis for a detailed description of the method). The average ICC over all seed networks was fair at $ICC = 0.46$ $(0.073SD)$. The range of the within-sample stability of continuous assignments to FC subtypes across networks was $ICC = 0.3$ for the amygdala-hippocampal complex network to $ICC = 0.63$ for the lateral default mode network (*Figure 2*).

We next evaluated the impact of the amount of available data on ICCs. For this we repeated the analysis in Longitudinal sample 2, a separate, general population dataset, wherein 10 scan sessions were available for each individual. We again computed FC subtypes on one scan session. From the remaining sessions, we then took two sets of 1, 2, 3, and 4 sessions, and computed continuous assignments to the FC subtypes based on seed-FC data averaged across the sessions (i.e. across data from 1 to 4 sessions) in each set (see *Figure 1*). In this data set, the average ICC of continuous assignments to FC subtypes was fair at $ICC = 0.57$ $(0.094)$ when each assignment was computed on a single session.

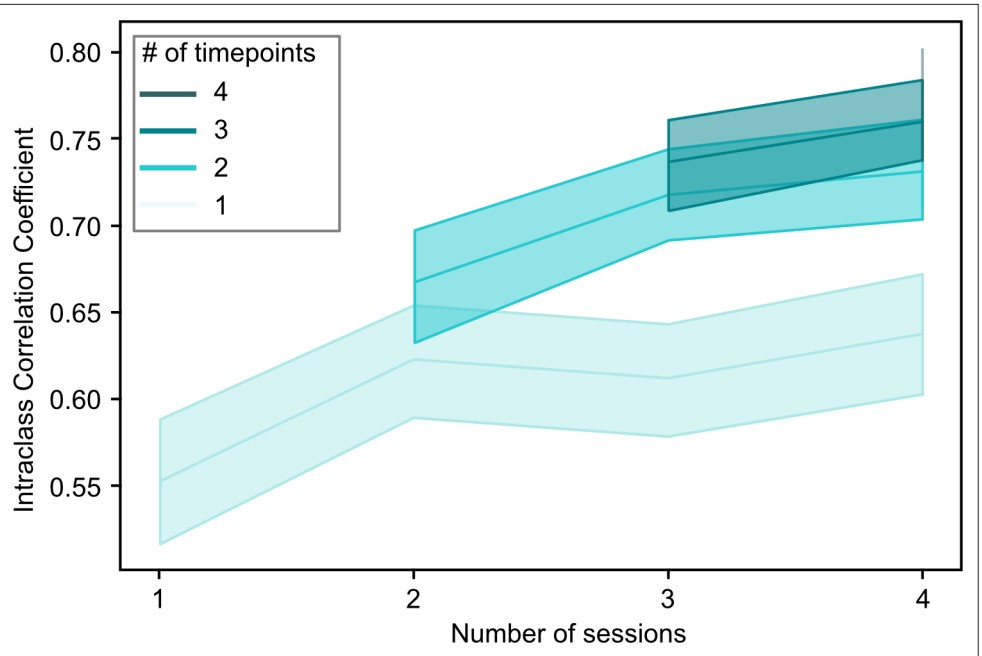

**Figure 3.** Overview of the relationship between the stability of continuous subtype assignments (Intraclass Correlation Coefficients) and the amount of data used to compute these assignments. Continuous subtype assignments were computed on samples in longitudinal sample 2 (n=26, 10 sessions per participant) and for pairs of single scan sessions, or for pairs of averages of multiple scan sessions (2–4, horizontal axis). We also controlled the total number of time points included in the averages to be the length of a single (light green) or multiple sessions (darker green hues). Opaque lines represent the average ICC across repeated samples of session pairs, and shaded areas reflect the 95% confidence interval. This analysis allowed us to investigate whether changes in stability were driven more by the inclusion of multiple sessions, or by the total amount of timepoints included in the average.

When estimating continuous assignments on the average of multiple sessions, the ICC increased markedly: good ($ICC = 0.68, SD = 0.09$) for 2 sessions, good ($ICC = 0.75, SD = 0.071$) for 3 sessions, and excellent ($ICC = 0.80, SD = 0.067$) for 4 sessions.

By combining data from multiple scan sessions to compute the continuous assignments, we increased both the total length of available scan data, and the number of sessions. In order to better understand whether the observed increase in stability of continuous FC subtype assignments was driven by the total amount of data or the number of combined sessions, we repeated the analysis but kept the number of included time frames constant. For example, for a session length of 230 time frames, we would take 230/2=115 time frames for an average of 2 sessions, 230/3=60 time frames for an average of 3 sessions and so on. We found that ICC of continuous assignments to FC subtypes increased both as a function of the number of included scan sessions and also as a function of the total amount of available time frames (***Figure 3***).

Finally, we evaluated the stability of continuous assignments to FC subtypes that were computed on independent data (out of sample stability). To this end, we computed FC subtypes on the discovery sample and estimated continuous assignments to these FC subtypes for individuals in the Longitudinal sample 1, a mixed patient-control sample (***Figure 1***). Here the average ICC was poor at $ICC = 0.33$ ($SD 0.072$) with a range of $ICC = 0.23$ in the inferior temporal gyrus to $ICC = 0.48$ in the medial ventral attention network (***Figure 2***). We thus showed that the stability of continuous assignments to FC subtypes ranges from poor to excellent as a function of the amount of available data per individual and whether FC subtypes and continuous assignments are computed on the same data.

## FC subtypes show association with ASD diagnosis

We next investigated whether any of the FC subtypes identified in the discovery sample, naturally captured interindividual variance related to clinical ASD diagnosis. To test this question, we computed

the continuous assignment of all individuals in the discovery dataset to the identified FC subtypes (see Association with autism diagnosis). We then tested for a linear relationship between continuous assignments to these FC subtypes and ASD diagnosis (i.e. ASD or NTC).

We identified 11 FC subtypes for which the continuous assignment of individuals were significantly associated with the clinical diagnosis of ASD, after correction for multiple comparisons ($p_{adj}$ reflects the false discovery rate adjusted p-values, see Materials and methods). That is, ASD and NTC individuals differed significantly in their continuous assignments with these FC subtypes.

NTC individuals showed significantly stronger continuous assignments than ASD individuals with 5 of the 11 FC subtypes (i.e. the association with ASD diagnosis was negative). These negatively associated (negASD) FC subtypes originated from seed networks in the ventral motor network ($T = 3.79, p_{adj} = 0.0037, d = -0.42$) and dorsal motor network ($T = 4.15, p_{adj} = 0.0018, d = -0.49$), the auditory network ($T = 4.25, p_{adj} = 0.0018, d = -0.52$), the medial ventral attention network ($T = 3.49, p_{adj} = 0.0091, d = -0.39$), and the downstream visual network ($T = 3.23, p_{adj} = 0.0196, d = -0.38$). An overview of the spatial pattern of above- and below-average FC among the negASD FC subtype maps is shown in *Figure 4* (see also Functional connectivity subtypes associate robustly with ASD diagnosis for a topographic overview of all identified FC subtypes). The negASD FC subtype maps share a number of characteristics, despite originating from different seed networks: they are characterized by above average FC within each seed network (seed network denoted by a bright green outline); they show above average FC in overlapping brain regions, particularly within sensorimotor areas around the central sulcus, visual areas in the occipital cortex, auditory areas along the superior temporal gyri and opercula; and they show below average FC in overlapping brain regions, particularly within areas of the ventromedial prefrontal cortex and subcortical areas involving the basal ganglia and thalamus.

ASD individuals showed significantly stronger continuous assignments than NTC individuals with 6 of the 11 FC subtypes (i.e. the association with ASD diagnosis was positive). These positively associated (posASD) FC subtypes originated from seed networks in the ventral motor network ($T = 2.91, p_{adj} = 0.0330, d = 0.32$), the dorsal motor network ($T = 3.8-, p_{adj} = 0.0369, d = 0.39$), the downstream visual network ($T = 2.94, p_{adj} = 0.0330, d = 0.28$), the amygdala-hippocampal complex ($T = 2.75, p_{adj} = 0.0488, d = 0.27$), the fronto-parietal control network ($T = 2.92, p_{adj} = 0.0330, d = 0.29$), and the lateral default mode network ($T = 3.17, p_{adj} = 0.0204, d = 0.30$). An overview of the spatial pattern among the posASD FC subtype maps is shown in *Figure 4*. The posASD FC subtype maps also share a number of characteristics, despite originating from different seed networks: they generally show below average FC within each seed network and they show an overal pattern of below average FC throughout the brain. Compared to negASD FC subtype maps, posASD FC subtype maps appear more heterogeneous in their profile of FC alteration. We thus showed that a subset of the identified FC subtypes naturally captured some inter-individual variation of the clinical ASD diagnosis.

## Spatial topography of ASD FC subtypes are consistent when cerebellum is included

Due to the limited coverage of the cerebellum in our datasets, cerebellar seed regions were excluded from our main analysis (see Quality control of imaging data). However, FC alterations involving the cerebellum have been repeatedly reported in the ASD literature (*Lake et al., 2019*; *Oldehinkel et al., 2019*; *Sathyanesan et al., 2019*). We therefore repeated our analysis among a subset of individuals with complete brain coverage in the discovery and replication samples and included seed FC maps of the two cerebellar seed regions of the MIST_20 brain atlas in the analysis *Figure 5*. We identified three FC subtypes that were negatively associated with ASD in the discovery sample. These three negASD FC subtypes originated from seed regions in ventral and dorsal sensorimotor network and in the auditory seed network. All three negASD FC subtypes identified in the discovery sample were also positively associated with ASD diagnosis, although with reduced effect sizes ($d_{discovery} = 0.45$ vs $d_{replication} = 0.13$). No cerebellar FC subtypes were found to be significantly associated with ASD diagnosis.

All three negASD FC subtypes shared a similar spatial topography that was characterized by above average FC with sensorimotor and auditory brain regions, and below average FC with subcortical and cerebellar regions, as well as areas in the ventromedial prefrontal cortex. In general, the spatial topography of the negASD FC subtypes identified in the subsample with full brain coverage (including the

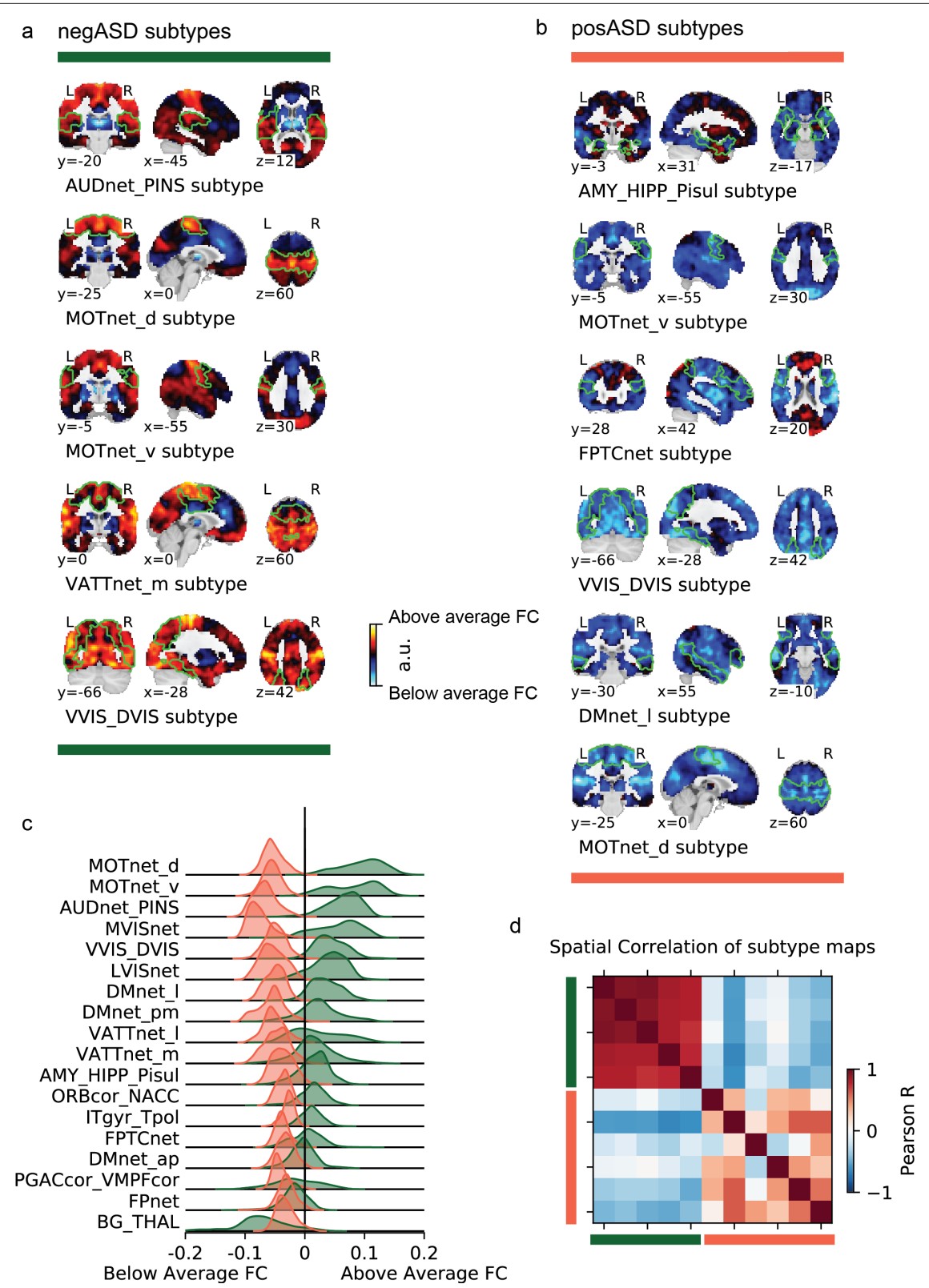

**Figure 4.** Overview of negASD and posASD FC-subtype maps. Maps of negASD (**a**) and posASD (**b**) FC-subtype (corresponding seed networks are outlined with a thin green boundary on the map). MIST_20 seed network names are abbreviated, see *Table 1* for full seed network names. (**c**) Decomposition of the average negASD (green) and posASD (red) FC-subtype map into 18 brain networks. (**d**) Spatial correlation between FC-subtype maps. negASD (green) and posASD (red) FC-subtypes are denoted by colored bars along the correlation matrix.

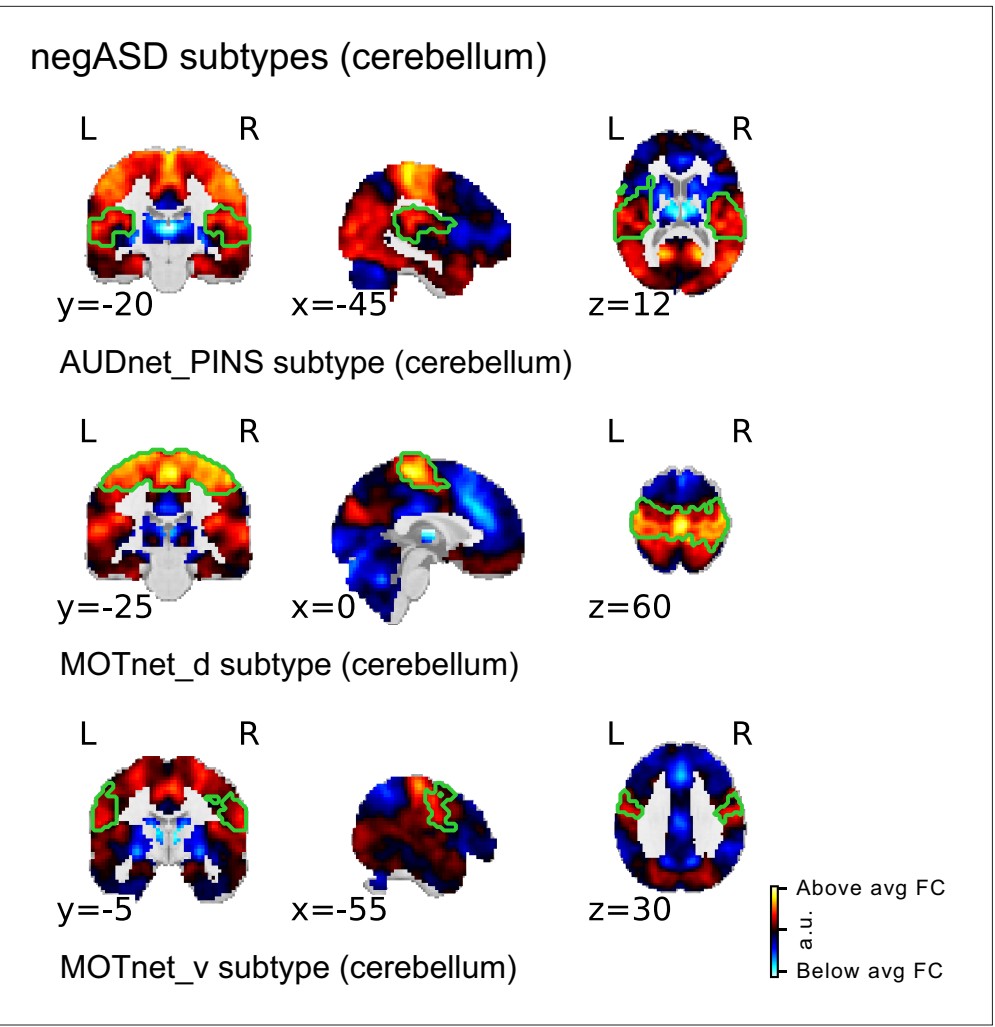

**Figure 5.** Spatial maps of subtypes with significant association to ASD diagnosis for supplementary analysis with cerebellar coverage. The main subtype and ASD association analysis was repeated in a subsample of the discovery sample with sufficient cerebellar coverage to include all 20 MIST_20 seed networks (including 2 cerebellar seed regions). All significant ASD diagnosis associations were negative (i.e. negASD subtypes). MIST_20 seed network names are abbreviated, see *Table 1* for full seed network names.

cerebellum) was very similar to the shared topography among negASD FC subtypes identified in the main analysis (excluding the cerebellum). We thus concluded that the exclusion of the cerebellum in our main analysis did not change our central findings.

## FC subtypes show limited association with ASD symptom measures beyond diagnosis

We next investigated whether any FC subtypes captured interindividual variation of ASD related symptoms. To do so, we tested for linear associations between continuous assignments to FC subtypes and total scores of the Autism Diagnostic Observation Schedule (ADOS, *Gotham et al., 2007*) as well as total scores of the Social Responsiveness Scale (SRS, *Constantino and Gruber, 2002*). Because the current clinical best practice to diagnose Autism relies on ADOS as a diagnostic instrument, ADOS scores are highly correlated with an Autism diagnosis. We therefore confined our investigation to individuals with an ASD diagnosis to determine whether any FC subtypes exhibited an association with ADOS scores that went beyond the identified effects of the ASD diagnosis. We did not observe any significant association between total ADOS scores and continuous assignments to FC subtypes beyond the effect of ASD diagnosis.

It is possible that the effect of ASD diagnosis on seed FC maps is dominating the identification of FC subtypes and thus masking more subtle links between FC subtypes and ASD symptoms that only exist among individuals with a diagnosis of ASD. To test this possibility, we repeated the FC subtype analysis only among ASD individuals in the discovery dataset ($N_{ASD} = 278$) and then tested for linear associations between total ADOS scores and continuous assignments of individuals to the FC subtypes identified in this subsample. We did not observe any significant association between total ADOS scores and continuous FC subtype assignments in the subsample of ASD individuals in the discovery dataset.

Lastly, we also tested for the association between total SRS scores and continuous assignments to FC subtypes identified in the discovery sample ($N = 388, N_{ASD} = 194$). Compared to the ADOS, the SRS is a shorter screening tool for characteristic, ASD-related social behavioral patterns and thus SRS scores are available for some NTC individuals in our sample ($N = 199, N_{ASD} = 108$). We therefore tested for a linear association between continuous assignments to FC subtypes and SRS scores in the mixed NTC and ASD sample for whom SRS scores were available.

We did find one FC subtype based on the ventro-medial prefrontal seed region that showed a significant association with raw SRS scores ($r = 0.24, p_{adjust} = 0.04$) after correcting for multiple comparisons. This FC subtype was also significantly associated with ASD diagnosis and showed the strongest effect size of all posASD FC subtypes (cohens-d=0.38). On the replication sample, the association of this FC subtype with SRS scores was no longer significant after correction for multiple comparisons, but still showed an effect in the same direction ($r = 0.17, p_{adjust} = 0.06$).

## negASD and posASD FC subtypes show similar spatial patterns of FC alterations

We noticed that the spatial pattern of negASD FC subtype maps appeared similar, despite representing connectivity profiles from different seed networks (*Figure 4a and b*). Similarly, the spatial maps of posASD FC subtypes all appeared to show below average connectivity. We therefore investigated whether negASD or posASD FC subtypes shared similar FC profiles and whether this similarity also extended to the continuous assignments of individuals to these FC subtypes. We found that negASD FC subtypes exhibited a highly convergent pattern of FC alterations ($\tilde{r}_{spatial} = 0.81$, where $\tilde{r}$ reflects the median spatial correlation across FC subtype pairs) that was distinct from those of posASD FC subtypes ($\tilde{r}_{spatial} = -0.3$). The spatial similarity among posASD FC subtypes was less pronounced ($\tilde{r}_{spatial} = 0.3$) than that of negASD FC subtypes. This finding extended to continuous assignments that were more strongly correlated among negASD FC subtypes ($\tilde{r} = 0.62$) than among posASD FC subtypes ($\tilde{r} = 0.21$), and anti-correlated between negASD and posASD FC subtypes ($\tilde{r} = -0.25$). By dividing all FC subtype maps into the 18 seed networks, we observed that the shared spatial pattern of negASD FC subtypes was characterized by overconnectivity with unimodal sensory brain networks, and underconnectivity with the basal ganglia and fronto-parietal network (green hues, *Figure 4c*). By contrast, the shared spatial pattern of posASD FC subtypes was characterized by pervasive underconnectivity (red hues, *Figure 4c*). We thus showed that negASD and posASD FC subtypes exhibited similarities of FC alteration and continuous assignments, and that these similarities were more pronounced for negASD FC subtypes than posASD FC subtypes.

## Differences in whole-brain connectivity contribute to FC subtypes

Because of the striking spatial similarity of negASD FC subtypes we investigated whether the identified FC subtypes were driven by in whole-brain FC. Within the discovery sample, and for each seed network separately, we regressed the individual whole-brain FC (together with other covariates of non-interest, see Functional connectivity estimation for details) from the seed FC maps before repeating the FC subtype extraction process and computing the association with ASD diagnosis. We found seven FC subtypes that originated from primary sensory seed networks seeds in the motor, auditory and visual networks, (see *Figure 6*) and were significantly associated with ASD diagnosis. Three of the FC subtypes were negatively associated with ASD (negASD) and showed strong topographic similarity of their FC subtype maps that very closely resembled the previously identified profile of negASD FC subtypes not corrected for whole-brain FC. This negASD pattern was characterized by above-average FC in brain regions associated with primary sensory functions such as motor, visual, and auditory regions, and by below average connectivity with medial-prefrontal and subcortical regions. The

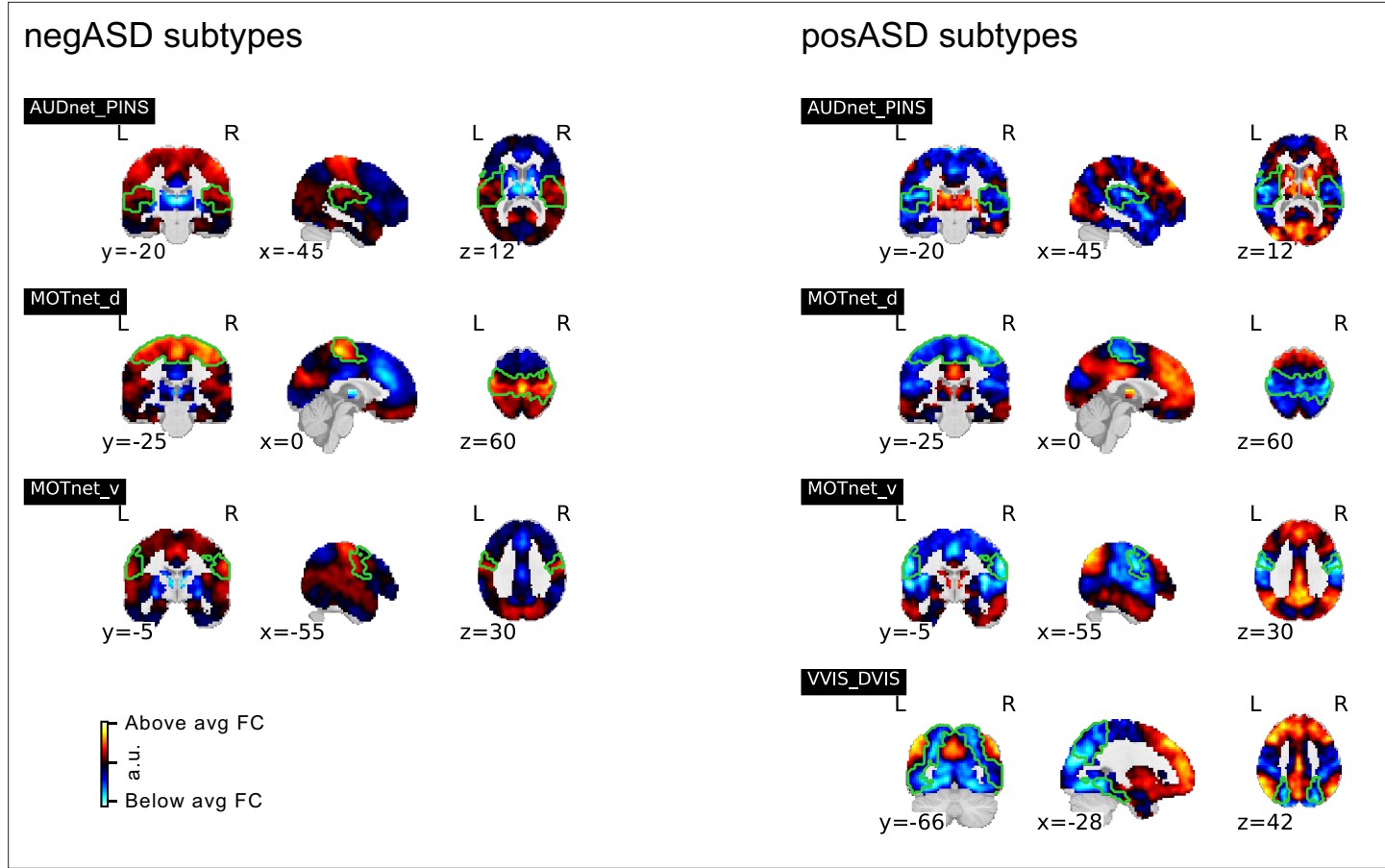

**Figure 6.** Spatial maps of subtypes with significant association to ASD diagnosis for supplementary analysis that regressed global connectivity from all seed FC maps. Subtypes with significant negative association of continuous assignments and ASD diagnosis are shown on the left, those with positive associations on the right. MIST_20 seed network names are abbreviated, see *Table 1* for full seed network names.

remaining four FC subtypes were positively associated with ASD (posASD) and, in contrast to posASD FC subtypes not corrected for whole-brain FC, showed topographic similarity of their FC subtypes maps. This posASD pattern was the inverse of the negASD pattern, and thus characterized by below average FC in sensory brain areas and above average FC in subcortical and ventromedial regions, and in the precuneus.

## Associations between FC subtypes and ASD diagnosis replicate moderately

We next investigated how reproducible the discovered association between FC subtypes and ASD diagnosis was in an independent dataset. For each of the FC subtypes that showed a significant association with ASD diagnosis in the discovery dataset, we computed the continuous assignment for the individuals in the independent replication dataset. In this way, we tested the out of sample reproducibility of the observed association effect. We tested different degrees of replication: whether the observed effect in the replication sample was significant after correction for multiple comparisons, significant at an uncorrected $p < 0.05$, whether the estimated magnitude of the effect fell within the 90% confidence interval of the effect size estimate in the discovery sample, and whether the effect in the replication sample had the same direction as the one estimated in the discovery sample. We found that all effects of association with diagnosis in the replication sample had the same direction as in the discovery sample. The ASD subtype effect size estimates in the discovery sample were correlated at $r = 0.91$ ($r = 0.71$ across all 87 subtypes) with the effect size estimates in the replication sample. The magnitude of the estimated effects in the replication sample was on average 63% of those estimated in the discovery sample, and nine out of eleven

effect size estimates fell within the 90% confidence intervals of the effect size estimates in the discovery sample. Five of those effects were significant at $p < 0.05$, and two of those were significant at $p_{adj} < 0.05$ (**Figure 4b**). We thus showed that the association between FC subtypes and ASD diagnosis observed on the discovery dataset was moderately replicable in the independent replication dataset.

## Discussion

ASD is characterized by a heterogeneity of symptoms and neurobiological endophenotypes (**Nunes et al., 2019**; **Dickie et al., 2018**; **Jacob et al., 2019**) among the affected individuals. Data driven, unsupervised clustering appears as a natural approach to decompose the heterogeneity in ASD and to identify FC subtypes of functional brain connectivity. Here, we first sought to evaluate how stable and reproducible FC subtypes are when derived from a heterogeneous sample of both neurotypical and autistic individuals. We then investigate whether fully data driven FC subtypes are associated with a clinical diagnosis of ASD. Our results suggest that data driven FC subtypes are moderately reliable on currently available datasets and show a weak to moderate association with the clinical diagnosis of ASD, that generalizes to independent replication data.

### Functional connectivity subtypes are stable

Our results shed light on previous findings on the robustness (**Easson et al., 2019**) or non-reproducibility (**Dinga et al., 2019**) of subtype analyses. The spatial patterns of the FC subtypes were found to be robust to perturbations in the discovery data set. This observation fits with previous studies that reported stable imaging subtypes in ASD (**Hong et al., 2018**; **Easson et al., 2019**; **Tang et al., 2020**). By contrast, we found that making a discrete assignment of individuals to FC subtypes was not robust to perturbations. Our findings are consistent with the study of **Dinga et al., 2019** that could not replicate discrete subtypes originally described in depressed patients (**Drysdale et al., 2017**) using an independent dataset. Dinga and coll. concluded that the data instead supported a more parsimonious model of continuous neurobiological axes.

In the wider ASD literature, the robustness of discrete subtype assignments has been more comprehensively investigated for symptom based subtypes. Several symptom based subtypes of autism have been proposed in attempts to provide more homogeneous diagnostic criteria. However, the distinction between these subtypes was also not found to be well supported by replication attempts which has led the field to merge sub-diagnoses of autism under the label of autism spectrum disorder (**Lord et al., 2012**; **Volkmar and McPartland, 2014**). In line with these observations, our results suggest that the assignement of an individual to an FC subtype is better constructed as continuous, rather than discrete. For individuals who are equally similar to two different FC subtypes, a small change of the connectivity profile may be enough to change a discrete method. By contrast, the continuous similarity measure would not change drastically. An emerging body of literature therefore conceptualizes subtypes as latent dimensions that can be expressed to varying degrees in each individual (**Kernbach et al., 2018**; **Tang et al., 2020**; **Easson et al., 2019**). Our own results support this view: we find that, unlike discrete assignments to FC subtypes, continuous measures of an individuals' similarity with each FC subtype are moderately robust and can be very robust when more data is available per individual to compute the continuous assignment. The ICC of continuous assignments to FC subtypes computed on separate data was low but consistent with previous reports of the robustness of single session seed based FC measures (**Shehzad et al., 2009**; **Birn et al., 2013**; **Noble et al., 2019**).

When the continuous assignments to FC subtypes were computed based on the average FC of multiple scan sessions per individual, we found high to very high stability measures that were in line with the well-established link between scan length and FC reliability (**Gordon et al., 2017**). We further clarified this relationship by showing that both the number of averaged scan sessions and also the total number of included time points across sessions contribute to the observed increase of ICC, although scan duration leads to markedly higher gains in stability. The generalizability of the associations between continuous assignments and clinical ASD diagnosis may thus increase if longer or repeated scan sessions were available for the replication sample.

## Moderate and reproducible association between FC subtypes and ASD diagnosis

The majority of previous subtyping analyses in ASD have been constrained to patients who were already diagnosed with ASD (*Hrdlicka et al., 2005*; *Hong et al., 2018*; *Tang et al., 2020*). We have instead used an unsupervised clustering approach to identify diagnosis-naive FC subtypes across autistic and neurotypical individuals in order to determine whether they would associate with ASD diagnosis. Our results showed that these FC subtypes were significantly associated with a clinical diagnosis of ASD, and that the observed effects were small to moderate, ranging between $d = 0.3$ and $d = 0.5$ (or from $r = 0.15$ to $r = 0.24$ when expressed as a correlation coefficient) on the discovery sample, with reduced effect sizes identified in an independent replication sample.

Our effect sizes are comparable to those reported by other imaging based subtypes in ASD, which have all estimated association with diagnosis in their discovery sample (i.e. have not been replicated on independent data). A recent study (*Kernbach et al., 2018*) investigated the heterogeneity of FC in mixed data of ASD, NTC, and attention deficit hyperactivity disorder (ADHD), a common comorbidity of ASD individuals (*Rommelse et al., 2010*). The authors identified one FC endophenotype that was weakly associated with ASD ($r = 0.15$) but extended both to ADHD and NTC individuals. The magnitude of the association between data driven FC subtypes and clinical diagnosis in our analyses is therefore comparable to what has been previously reported by other imaging based subtypes analyses of ASD. Weak-to-moderate associations between data driven FC subtypes and clinical diagnosis thus seem robust to the employed subtyping method, at least in the currently limited number of published studies.

With the exception of one FC subtype, we did not find significant associations between the identified FC subtypes and measures of ASD symptom severity such as the ADOS and SRS. This stands in some contrast to previous reports that have generally reported associations between imaging derived ASD subtypes and ADOS severity measures. One previous study among ASD patients (*Hong et al., 2018*) found that ADOS severity scores could be better predicted by a multivariate model on the basis of structural cortical alterations when individuals were first divided into three subtypes ($r = 0.47$ compared to $r = -0.12$ when the association was computed without regard for subtypes). A second study on FC derived imaging subtypes in a mixed sample of both ASD and NTC individuals (*Easson et al., 2019*) found that two whole-brain imaging subtypes best approximated the data, and that dividing individuals into these two subtypes allowed for multivariate relationships between FC, ADOS, and SRS scores (among others) to be revealed. Our own findings did not reveal similar relationships between FC profiles and symptom severity measures, even for a supplementary FC subtype analysis conducted only among ASD individuals to test whether such a relationship might have been obscured by the stronger diagnostic effect in the mixed sample. One explanation for the different findings is that we investigated a simple linear relationship between FC subtype assignments and ASD severity measures, whereas previous studies used subtyping findings to stratify their sample before conducting a multivariate model fit using all brain features. It is possible that similarly combining many FC subtype assignment scores in a multivariate prediction model would lead to stronger prediction of ASD severity.

Very few imaging based subtype analyses have been replicated on independent data, and to our knowledge none have so far been replicated successfully (*Dinga et al., 2019*). The replication of our results on independent data therefore establishes a novel benchmark of reliability for imaging based subtype analyses in ASD. We found that the observed effect sizes of the association between FC subtype assignments and clinical ASD diagnosis strongly correlated between the discovery dataset and the independent replication dataset ($r = 0.71$ across all FC subtypes, and $r = 0.91$ among those significantly associated with ASD in the discovery data), however effect sizes in the replication data were on average only 2/3 of the magnitude of those in the discovery data. This reduction of effect sizes on the replication data is expected as it reflects the inherent bias of significance testing to select larger effects and further underlines the importance of reporting original findings together with independent replications for an unbiased estimate (*Vul and Pashler, 2012*). Because no other imaging subtype analysis in ASD has been independently replicated to date, our results have to be interpreted in the context of replication attempts in the ASD case-control literature. The largest case-control analysis of FC alterations to date (*Holiga et al., 2019*) reported FC group differences between ASD and NTC individuals with effect sizes between $d = 0.46$ and $d = 0.6$, similar in size to our own results

of FC subtype associations with clinical ASD diagnosis. Using several large replication samples, the authors then showed that these results were reproducible in independent data, however with similarly depressed effect sizes (i.e. $d \approx 0.2$).

## FC subtypes converge on a single dimension of ASD related variation

In our analyses, we hypothesized that FC subtypes linked to different seed networks might be linked to distinct behavioral ASD symptom profiles. This hypothesis was not supported by our findings. The converging topography among the identified ASD-related FC subtypes rather suggests that all FC subtypes capture different views of a single underlying FC profile characterized by a distinction between unimodal and transmodal FC, with negASD and posASD FC subtypes situated at opposite ends of a shared dimension. When we corrected for individual differences in average global FC, we identified posASD FC subtypes that showed a topography of FC alteration opposite of that shown by negASD FC subtypes: below average FC within and between unimodal brain networks, and above average FC with transmodal regions. We confirmed this observation by a supplemental principal component analysis (*Appendix 3—figure 1*). Across all 18 seed networks, the components explaining the largest amount of FC variance captured whole-brain shifts in FC. The spatial pattern is strikingly similar to the case-control Holiga map (*Figure 7*). We computed the spatial correlation between the two patterns at $r = -0.6$, as high or higher than the reported replicability of the case-control pattern itself (between $r = 0.3$ and $r = 0.6$, depending on the replication sample). The specific pattern of connectivity differences is also very much in line with a view of the connectome as organized along major gradients, the most dominant of which parses the brain from sensori-to-associative. Our results here can then be interpreted as a compression of the gradient (less segregation) in ASD, which is a similar conclusion to a recent gradient-based analysis in ASD, also in ABIDE (*Hong et al., 2019a*). This observation was not an assumption made by our model, but rather a conclusion reached by parsing many seed regions, and stands opposite to our initial hypothesis. If granular and symptom-specific FC subtypes do exist, they are dominated by larger variations in gradients which would need to be taken into account. It is also possible that FC subtypes not associated with ASD may carry important, complementary information which would only become apparent in multivariate analysis combining different seeds, or with a larger sample with increased statistical power. The replication of association between FC subtypes and ASD between ABIDE 1 and ABIDE 2 indeed seemed to extend to many subtypes which were not significant after correction for multiple comparisons (*Figure 8*).

## Limitations

Our findings have to be interpreted in light of the limitations of the available data. First, our analyses only included male individuals which is a common problem in the field (*Khundrakpam et al., 2017*; *Hong et al., 2019a*) due to the higher frequency of ASD diagnosis among male individuals (*Lai et al., 2014*). Recent data curation efforts have therefore started to deliberately include more female individuals (*Di Martino et al., 2017*; *Bedford et al., 2020*).

Inconsistent coverage of the cerebellum led us to exclude the cerebellum from our main analyses. Cortico-cerebellar functional connectivity has repeatedly been shown to be altered in individuals with ASD, and these FC alterations have been linked to symptom severity. We have conducted a supplemental analysis among a subset of individuals with complete cerebellar coverage to investigate whether this inclusion would be likely to alter our main findings. In this subsample, we identified FC subtypes with very similar spatial topography to our main findings, suggesting that our results would not have changed substantially because of the inclusion of the cerebebellum.

The distribution of age among individuals in our discovery and replication sample was different, despite identical sampling criteria. Age is an important covariate to consider, particularly in a developmental disorder like ASD. It would have been possible to match the age distributions but this would have resulted in a drastically reduced sample size. We still observed a good replication of the association between FC-subtypes and ASD.

As we previously discussed, our findings are limited by the amount of data available per individual, and in particular regarding the stability of FC subtype assignments. A promising direction for future research will be to investigate ASD datasets with longer time series (*Allen et al., 2014*), which would allow researchers to incorporate dynamic FC subtypes in ASD. Also note that there is a limitation in our estimation of subtype reliability. An individual may contribute to a subtype map, hence inflating

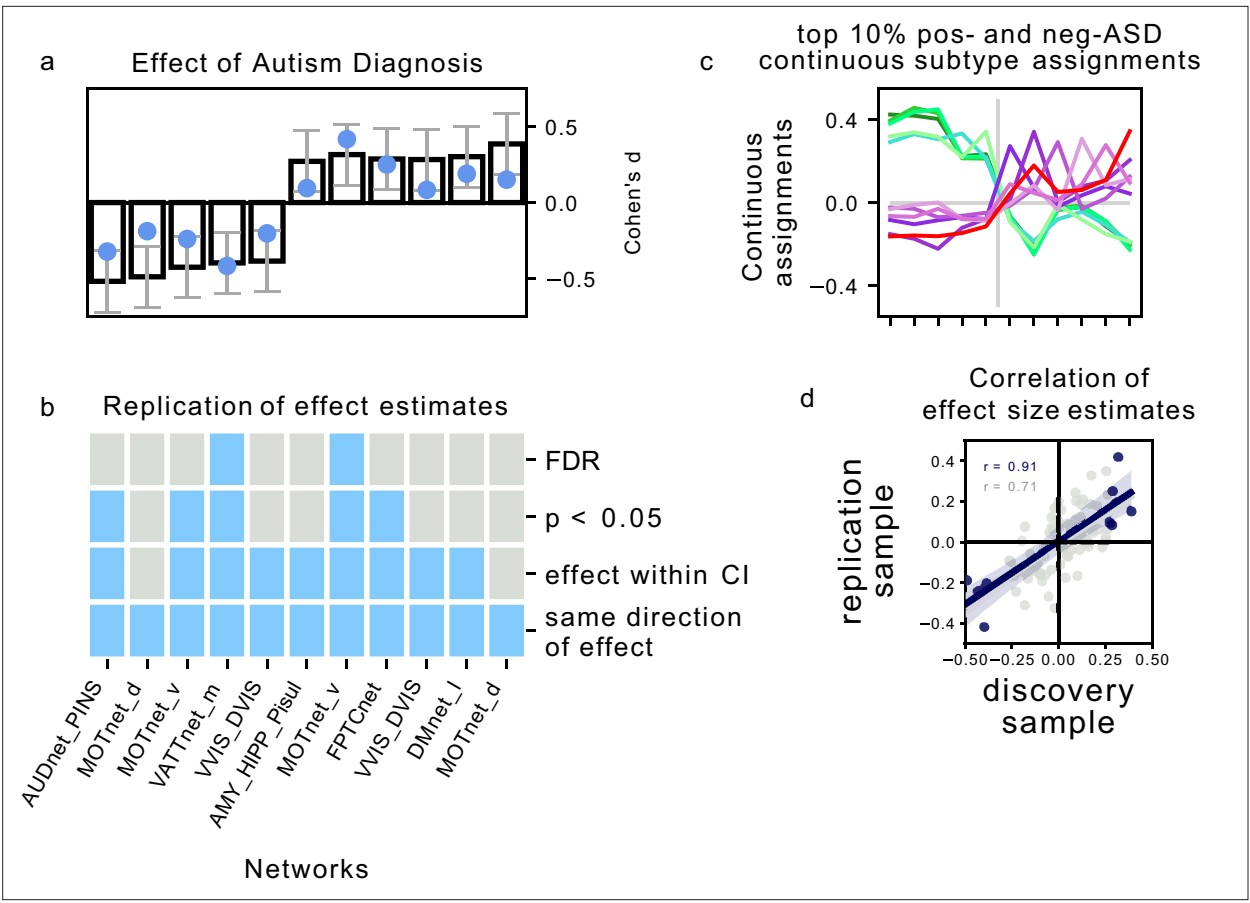

**Figure 7.** Association of continuous assignments to FC-subtypes and diagnosis. (**a**) Bar plots represent the standardized group difference (Cohen's d) of continuous assignments to FC-subtypes between NTC individuals and ASD patients. Negative value reflect greater similarity of neurotypical control subjects with the FC-subtype, positive values reflect greater similarity of ASD patients with the FC-subtype. Error bars reflect the 95% confidence interval of the effect size estimates. The effect size observed in the independent replication data set is shown as a blue dot. (**b**) Matrix showing the degree of replication in the independent replication dataset of the observed association with diagnosis for each of the 5 negASD and 6 posASD FC-subtypes. Each row corresponds to a bar-plot in (a). From top to bottom, the degrees of replication are: FDR: full replication of the effect after FDR correction, $p < 0.05$: replication of the effect for uncorrected statistics, effect within CI: observed effect size in the replication sample falls within the 95% confidence interval of the observed effect in the discovery sample, direction: observed effects in the discovery and independent replication sample go in the same direction. (**c**) Graph illustrating the similarity of continuous assignments to posASD and negASD FC-subtypes. The average continuous assignments of the top 10% of individuals with the highest similarity with a negASD (green shades) or posASD (red shades) FC-subtype are displayed across all identified negASD (left side) and posASD (right side) FC-subtypes. Note that an individual may belong to the top 10% in more than one FC-subtype, and we did check that the conclusions are robust for other thresholds (5%, 15%). (**d**) Correlation plot of the observed effect sizes in the discovery and independent replication datasets. The dark blue line represents the correlation of effect sizes for subtypes with significant ASD associations in the discovery sample ($r = 0.91$), the light grey line for all 87 subtypes ($r = 0.71$). The grey shaded areas reflects the respective estimated 95% CI of the linear fit. MIST_20 seed network names are abbreviated, see **Table 1** for full seed network names.

assignment to this subtype. This limitation does not apply when we generate subtype maps from an independent sample from the individual, as we did in the replication analysis of the association with ASD.

Our results have focused only on individuals with ASD and NTC. Given the extensive evidence of overlap of symptoms (**Grzadzinski et al., 2011**) and neurobiological phenotypes between ASD and other neurodevelopmental disorders (**Sha et al., 2019**), a fruitful avenue for future research will be to extend this approach to investigate cross-diagnostic clusters of FC subtypes (**Elliott et al., 2018**).

We are re-using the same datasets as most other studies do too. This is a problem called dataset decay (**Thompson et al., 2020**). This repeated re-analysis of ABIDE may explain at least in part the convergence in results between our study and the study on gradient compression in ASD (**Hong et al., 2019a**), as well as the recent multi-study analysis on ASD case-controls (**Holiga et al., 2019**).

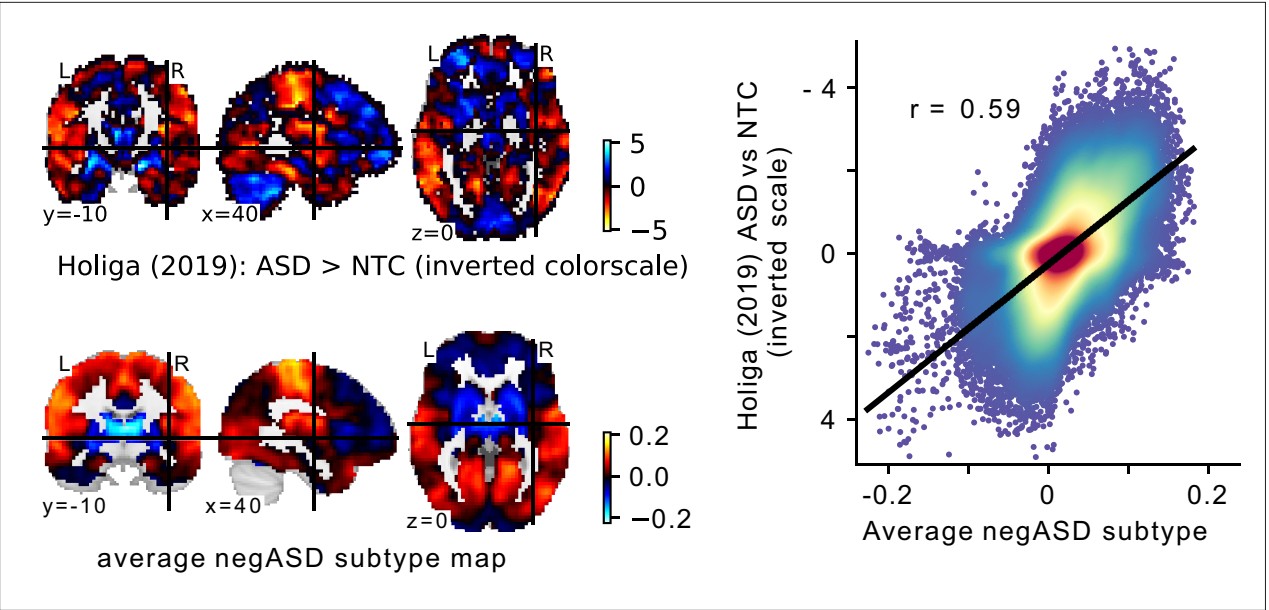

**Figure 8.** Comparison of the average negASD FC-subtype map to a case-control signature. (**a**) The spatial map of a large sample size case-control contrast between ASD and NTC individuals (top row), compared to the average spatial map of the negASD FC-subtypes identified on our data (bottom row). Note that because of the opposite nature of the two contrasts (i.e. ASD >NTC for the case-control contrast and NTC >ASD for the negASD FC-subtype map), the color scale for the case-control map has been inverted for better comparability. (**b**) Plot of the voxel-wise spatial correlation between the (inverted) case-control contrast map and the average negASD FC-subtype map. The blue to red color gradient reflects the density of voxels represented in each area of the graph.

## Conclusions

Our findings suggest that a basic cluster analysis leads to FC subtypes that are associated with clinical diagnosis and SRS. The stability of subtypes was greater for continuous than discrete assignments, and long time series led to higher stability which may prove important for precision medicine applications. The association between FC subtypes and clinical diagnosis was strikingly replicable on an independent sample. We systematically probed FC subtypes associated with ASD across the full brain and found a convergent topography, evocative of a compression of the primary gradient of functional brain organization. The low-to-moderate effect size of the observed association with ASD makes it clear that FC subtypes will not replace clinical categories in the future. Instead, our results support the use of FC subtypes to summarize high dimensional, heterogeneous data while retaining clinically meaningful variations.

## Materials and methods

### Discovery sample

The discovery sample consisted of imaging data from the ABIDE 1 dataset ($N = 388$, $N_{ASD} = 194$, $Age = 17.04$, (7.08), from 7 recording sites) and ASD individuals were matched with neurotypical controls on age ($Age_{ASD} = 17.0$, (7.28)) and head motion ($FD_{ASD} = 0.17mm$, (0.048)). The full ABIDE 1 dataset includes 1112 individuals from 20 imaging sites ($N_{ASD} = 539$, $age = 17.04$, (8.04)) of which 948 are male. Due to the strong sex imbalance of the data, we limited our analysis to male individuals. After preprocessing of the imaging data, 557 individuals ($278 ASD$, $Age = 16.65$, (6.75)) from 13 imaging sites were found to pass our quality control criteria. We then matched the NTC and ASD individuals at each site by age and head motion through propensity score matching without replacement (**Rosenbaum and Rubin, 1985**). The matched sample included 478 individuals ($239 ASD$, $Age = 16.67$, (6.67)). We further excluded the five imaging sites with fewer than 20 matched individuals, leaving 388 individuals for the final discovery sample.

A large number of individuals in the ABIDE 1 dataset had incomplete functional imaging coverage of the cerebellum (see Quality control of imaging data). We thus also selected the subset of male

individuals with complete cerebellar coverage for a separate analysis. After analogous selection propensity score matching to the main discovery sample we chose to retain sites with at least 10 matched individuals. The total sample of individuals with cerebellar coverage in the discovery sample consisted of 344 individuals ($N_{ASD} = 172$; $Age_{ASD} = 15.03, (5.72)$) from 11 recording sites.

## Replication sample

The replication sample consisted of imaging data from the ABIDE 2 dataset ($N = 300, N_{ASD} = 150$, from 7 imaging sites) and ASD individuals were matched with neurotypical controls on age ($Age_{ASD} = 12.0, (4.05)$) and head motion ($FD_{ASD} = 0.17, (0.053)$). The full ABIDE 2 dataset includes 1114 individuals from 19 imaging sites ($N_{ASD} = 521, Age = 14.86, (9.16)$) of which 856 are male. Analogous to the discovery sample we limited our analysis to male subjects. After preprocessing of the imaging data, 587 individuals ($N_{ASD} = 273, Age = 13.94, (5.9)$) from 16 imaging sites were found to pass our quality control criteria. These individuals were then matched by age and head motion within each site through propensity score matching without replacement. The matched sample included 424 individuals ($N_{ASD} = 212, Age = 13.66, (5.25)$). We further excluded 9 imaging sites with fewer than 20 matched individuals, leaving 300 individuals for the final replication sample.

Similar to the discovery sample, we selected the subset of male individuals with complete cerebellar coverage for a separate analysis. We chose to retain sites with at least 10 matched individuals. The total sample of individuals with cerebellar coverage in the discovery sample consisted of 424 individuals ($N_{ASD} = 212$; $Age_{ASD} = 13.5, (4.97)$) from 16 recording sites.

## Longitudinal sample 1

The first longitudinal test sample was taken from a subset of individuals in ABIDE 2 for whom multiple scan sessions were available. In ABIDE 2, longitudinal imaging data are available for 168 individuals from 4 imaging sites ($N_{ASD} = 88, Age = 21.24, (15.45)$) of which 154 are male. Analogous to the discovery and replication sample, we limited our analysis to male individuals. After preprocessing of the imaging data, 84 individuals ($N_{ASD} = 42, Age = 14.58, (6.29)$) from three imaging sites were found to pass our quality control criteria. We selected the two imaging sites with the largest number of acceptable individuals (ABIDEII-OHSU_1 and ABIDEII-IP_1) and randomly selected individuals at each site to enforce equal sized groups of NTC and ASD. Where more than two acceptable imaging scans were available for an individual, the two scans with the lowest average head motion were selected. The final longitudinal test sample consisted of 68 individuals ($N_{ASD} = 34, Age = 13.46, (5.79)$) from two imaging sites.

## Longitudinal sample 2

The second longitudinal test sample consisted of individuals in the general population Hangzhou Normal University dataset (http://dx.doi.org/10.15387/fcp_indi.corr.hnu1) released by the consortium for reliability and reproducibility (*Zuo et al., 2014*). The final sample included 26 individuals ($N_{male} = 14, Age = 24.58, (2.45)$) that were each scanned 10 times at 3-day intervals over the course of a month. We selected the 26 individuals (out of a total of 30 available individuals) for which all resting state scans passed visual quality control.

## Clinical scores and symptom severity

The individuals from the ABIDE 1 and ABIDE 2 samples included in this study were diagnosed with ASD by expert clinicians based on either the Autism Diagnostic Observation Schedule (ADOS) (*Lord et al., 2000*; *Gotham et al., 2007*) or the Autism Diagnostic Interview - Revised (*Lord et al., 1994*). The ADOS provides a total sum of ratings of observation items in the ADOS subdomains that reflects the severity of observed symptoms but is primarily intended for diagnostic purposes. Calibrated ADOS severity scores have been proposed as a standardized, research appropriate measure of symptom severity that is comparable across ADOS modules and is less dependent on demographic factors such as age (*Gotham et al., 2009*). Few individuals in the discovery (N=109, 93 ASD) and replication (88 ASD) samples had calibrated ADOS severity scores. We therefore also investigated associations with the raw ADOS total scores that were available in larger numbers in the discovery ($N = 213, N_{ASD} = 182$) and replication ($N = 157, N_{ASD} = 148$) sample.

## Imaging data preprocessing

All imaging data were preprocessed with the NeuroImaging Analysis Kit (NIAK) version 1.13 (*Bellec et al., 2011*). The preprocessing pipeline was executed inside a Singularity (version 2.6.1) software container (*Kurtzer et al., 2017*) to facilitate the reproducibility of our findings. Preprocessing of the functional imaging data consisted of the following steps: Head motion between frames was corrected by affine realignment with a reference image (median image across frames). The magnitude of framewise head displacement (FD) was estimated from the time course of the affine realignment parameters (*Power et al., 2012*). The reference image was then coregistered into the MNI152 stereotaxic space (*Evans et al., 1994*) through an initial affine alignment with the individual anatomical T1 image and a subsequent, non-linear coregistration of the T1 image with the MNI template. A high-pass temporal filter (0.01 Hz) was fitted to the whole time series by discrete cosine transform to remove slow time drifts. Time frames with excessive head motion ($FD > 0.4mm$) were then censored by removing the affected frame, as well as the preceding and the two succeeding frames from the time series (*Power et al., 2012*). Nuisance covariates were then regressed from the remaining time points: the previously estimated discrete cosine basis functions, the average signal in conservative masks of the white matter and lateral ventricles, and the first principal components (accounting for 95% of variance) of the six rigid-body motion parameters and their squares (*Lund et al., 2006*; *Giove et al., 2009*).

## Quality control of imaging data

We controlled the quality of preprocessed data manually and through quantitative cut off values. Data were visually checked by a trained rater following a standardized QC protocol (*Benhajali et al., 2020*) with a structured QC tool (*Urchs et al., 2018*). Individuals were excluded for coregistration failure and for incomplete brain coverage in the functional data. Individuals were also excluded from the analysis if fewer than 50 time frames remained after motion censoring or if the average framewise displacement exceeded 0.3 mm. During the visual QC we noticed that a large number of individuals in both the discovery and replication sample had incomplete coverage of the cerebellum. For our main analysis we included all individuals and chose to remove all cerebellar networks. To assess the impact of excluding the cerebellum, we ran a supplemental analysis with cerebellar seed networks on a subset of individuals with complete brain coverage.

## Functional connectivity estimation

We estimated the seed based FC maps of 18 non-cerebellar seed networks defined in the MIST_20 functional brain atlas (*Urchs et al., 2017*) for our main analysis. In our supplementary analysis on cerebellar contributions we included all 20 networks of the MIST_20 atlas including the cerebellum (see Quality control of imaging data). The MIST_20 atlas represents large, spatially distributed subcomponents of canonical FC networks. The seed to voxel FC maps were estimated as the Pearson correlation between the average time series signal of a seed network and the time series of all gray matter voxels in the brain (excluding the cerebellum). Within each sample separately, the individual seed FC maps were centered to the mean of the whole sample (ASD and NTC) and known sources of variance of non-interest were regressed for each voxel at the group level: linear effects of age, head motion and imaging site. In a supplemental analysis we also included whole-brain FC in each individual seed FC map as a covariate of non-interest in our linear. As a consequence, the individual seed FC maps in each sample represented the residual variance around the group mean after accounting for these factors.

## Identification of FC subtypes

To identify communities of individuals with similar seed FC patterns we computed the spatial correlation of all pairs of subjects in the discovery sample, separately for each seed network. We expressed the dissimilarity between pairs of individual seed FC maps as the absolute value of 1 - their spatial correlation. The 18 subject by subject dissimilarity matrices (one per seed network) thus contained values between 0 (no dissimilarity or a spatial correlation of 1)–2 (perfect dissimilarity or a spatial correlation of –1) with 1 denoting no spatial relationship (a spatial correlation of 0).

For each seed network separately, we characterized communities of individuals with similar seed FC maps by hierarchical agglomerative clustering of the dissimilarity matrix for each seed network using the unweighted average distance linkage criterion (*Müllner, 2011*). We applied two criteria

for the identification of seed FC communities: (1) the average dissimilarity between seed FC maps in a community could not be greater than 1, and (2) the community had to have at least 20 members. This allowed for small subsets of individuals with distinct seed FC patterns to not be assigned to any communities. Assigning individuals to FC subtypes in this way is a discrete process and we therefore refer to these assignments as discrete FC subtype assignments.

Within each seed FC community, we estimated the average seed FC map across all community members. This map reflected the seed FC subtype shared by the community members and we refer to these maps as the FC subtype map.

Finally, we computed the spatial similarity of each individual in the discovery sample with the identified seed FC subtype by spatial correlation of the individual seed FC map with the corresponding seed FC subtype map. The estimated spatial correlation coefficient is a continuous measure of an individual's similarity with each of the FC subtypes and we therefore refer to it as a continuous FC subtype assignment. Each individual had continuous FC subtype assignments for each identified FC subtype, ranging from -1 (perfect anticorrelation of the individual and the seed FC subtype map) to +1 (perfect correlation of the individual and seed FC subtype map).

## Stability analysis

Before we investigated the three aspects of FC subtype (FC subtype maps, and discrete and continuous assignments) in detail, we wanted to determine the robustness of these metrics to perturbations of the discovery data. We used two approaches: (1) to determine the robustness of discrete FC subtype assignments and FC subtype maps, we conducted a stratified subsampling scheme on our discovery sample, (2) to determine the robustness of continuous FC subtype assignments, we computed the within subject stability of continuous FC subtype assignments across repeated scan sessions for individuals in the longitudinal sample.

We randomly selected 1000 stratified subsamples of half of our discovery sample while preserving the equal ratio of ASD patients and NTC. Within each subsample, we repeated the full FC subtype characterization procedure: group level regression of nuisance sources of variance, characterization of communities of similar residual seed FC maps, estimation of seed FC subtype maps. The number of unique pairs of subsamples was large ($\approx 500,000$) and there was considerable overlap of individuals between subsamples. Therefore, we randomly selected 1000 unique pairs of subsamples to estimate the robustness of the FC subtype community membership and FC subtype maps to perturbations in the data.

We determined the robustness of discrete FC subtype assignments by computing the similarity of the communities an individual was assigned to within two subsamples using the Dice coefficient (***Dice, 1945***). For each pair of subsamples A and B, we first identified the intersect of individuals (i.e. those individuals that were present in both subsamples). For each individual we then computed the Dice coefficient of the communities it was assigned to in sample A and sample B. The Dice coefficient here computes the ratio of twice the number of individuals shared between both communities over the total number of individuals in both communities. Thus, if all community neighbours of an individual in sample A were also community neighbors of that individual in sample B, then the Dice coefficient will be 1. Conversely, if none of the community neighbours of an individual in sample A were community members of that individual in sample B, then the Dice coefficient will be 0. We computed the average Dice coefficient across all individuals shared between a pair of subsamples.

We determined the robustness of the FC subtype maps by examining the spatial correlation of FC subtype maps extracted in each pair of subsamples. For each pair of subsamples A and B, we computed the spatial correlation of all FC subtype maps in sample A with all FC subtype maps in sample B. If FC subtype maps were robustly identified, then we would expect that for each FC subtype map in sample A we can find at least one FC subtype map in sample B that is very similar. We therefore searched (with replacement) for each FC subtype map in sample A the FC subtype map in sample B with the highest spatial correlation. Since the number of FC subtypes extracted in each subsample was determined by the data, we allowed for FC subtype maps in sample B to be a match for multiple FC subtype maps in sample A. We then took the average of the maximal spatial similarity between FC subtype maps of sample A and B as a measure of the robustness of the FC subtype maps.

We computed the robustness of the continuous FC subtype assignments as the intraclass correlation coefficient between repeated scan sessions of the same individual. We first investigated the

robustness of assignments to FC subtypes that had been identified on data from a separate scan session but of the same sample (within sample robustness). Using the longitudinal sample 1, we identified FC subtypes for each network on scan session 1, and computed seed based FC maps for all individuals on the remaining two scan sessions. Independently for each scanning session we then centered the seed FC maps to the group mean and regressed covariates of non-interest for each voxel. The residual seed FC maps were then used to compute the continuous FC subtype assignments for the FC subtypes identified on scan session 1. The replicability of these continuous FC subtype assignments across the two remaining scan sessions was then estimated with the intraclass correlation coefficient.

Using the longitudinal sample 2, we tested whether continuous FC subtype assignments were more robust if they were computed on larger amounts of data per individual. Again, we identified FC subtypes for each network on the first scan session and computed individual seed FC maps on the remaining nine scan sessions. Within each scan session, the individual seed FC maps were then centered to the group mean and nuisance covariates were regressed. Two residual seed FC maps were then computed for each individual by averaging across two non-overlapping sets of scan sessions. A given set included either 2, 3, and 4 scan sessions, in order to investigate the impact on the number of sessions on stability.

To differentiate whether changes in replicability of continuous FC subtype assignment were driven by the number of time frames used to compute the average FC maps, or by the number of imaging sessions included in these averages, we conducted a follow-up analysis. We again used the first of ten imaging sessions to identify FC subtypes for each network. Across the remaining nine sessions, the minimum number of time frames for any individual imaging session was $N_{frame\_min} = 238$, and we truncated all imaging sessions to 238 time frames. When computing the average individual FC maps across $M$ imaging sessions, we then held the total number of time frames included in the average constant to multiples $K$ of $N_{frame\_min}$ according to: $\frac{K}{M} * N_{frame\_min}$ where $K \leq M$. For example, we computed averages across 3 imaging sessions and truncated each individual imaging session to $K = 1; \frac{1}{3} * 238 = 79$ time frames per session (for a total of 237 time frames across sessions), $K = 2; \frac{2}{3} * 238 = 159$ time frames per session (for a total of 477 time frames) and $K = 3; \frac{3}{3} * 238 = 238$ time frames per session (for a total of 714 time frames), and so on. This allowed us to investigate both the effect of including a greater number of total time frames in each average (by increasing $K$) and the effect of including a greater number individual sessions (by increasing $M$ while holding $K$ constant).

Finally, we computed the out of sample robustness of continuous FC subtype assignments based on the repeated scan session in the longitudinal sample 1 with the FC subtypes identified on the complete discovery sample. Again, the robustness was measured with the intraclass correlation coefficient of continuous FC subtype assignments across scan sessions.

## Robustness of findings to changes in the FC subtype pipeline

We further explored the robustness of analysis to changes in the parameters of the FC subtype pipeline. We did not explicitly specify the number of FC subtypes for each seed network. Rather, we applied a threshold on the maximum dissimilarity within a FC subtype. This threshold implicitly sets the number of FC subtypes, based on the observed structure of the dissimilarity matrix between subjects. In order to understand how robust our findings were to changes in this parameter, we repeated all analysis steps (i.e. the identification of FC subtypes, the test for associations with ASD symptoms, and the generalization to the independent replication data) for different values of the maximum dissimilarity parameter. To measure the spatial similarity of FC subtype maps identified for different dissimilarity parameters we computed their pairwise spatial correlation. We then compared the number of identified FC subtypes and the observed associations with ASD symptoms and their generalization to independent data qualitatively.

## Association with autism diagnosis

We explored whether seed FC subtypes existed for which the presence of an autism diagnosis explained a significant amount of variance of the continuous FC subtype assignments. We tested this for each FC subtype by comparing the means of continuous FC subtype assignments between ASD individuals and NTC with a general linear model with diagnosis as the explanatory factor. As we had taken care to ensure equal sizes of individuals in both diagnostic categories, we did not

use a correction for unequal variances. The estimated p-values were corrected at a false discovery rate (FDR) of 5% across all FC subtypes using the Benjamini and Hochberg method (*Benjamini and Hochberg, 1995*). We report the standardized group difference (Cohen's d) between diagnosis and continuous FC subtype assignments as a measure of the effect size of the association with the clinical diagnosis. We continued investigating FC subtypes for which a significant difference of continuous FC subtype assignments between ASD patients and NTC was found in the discovery sample.

Within the set of FC subtypes that showed a significant association with ASD diagnosis we investigated whether spatial similarity with the FC subtype map explained additional variance of the severity in clinical symptoms. Because symptom severity and the clinical ASD diagnosis were highly correlated, and because healthy individuals had compressed or missing scores for most severity measures, we only tested this association in individuals with a diagnosis of ASD. We investigated the linear relationship between continuous FC subtype assignments and severity estimates for the calibrated ADOS severity scores (*Gotham et al., 2009*) and also for the raw ADOS total scores. We reported the correlation between symptom scores and continuous FC subtype assignments as a measure of the effect size of the association with symptom severity after correction for multiple comparisons using FDR.

It is possible that the potentially larger differences between NTC and ASD individuals in the heterogeneous discovery sample are obscuring more subtle FC variability among ASD individuals and that this variability may be related to ASD symptom severity. To investigate this possibility further, we conducted a supplemental FC subtype analysis only among ASD individuals in the discovery sample. FC subtypes and continuous FC subtype assignments were computed for the subset of ASD individuals in the discovery sample. We then computed the correlation between the ADOS symptom scores and the continuous FC subtype assignments.

## Principal component analysis of network FC

We compared our clustering based FC subtypes to a principal component analysis (PCA). For each seed network, and voxel by voxel, we centered the seed FC maps to the cohort mean and scaled them to unit variance. We then decomposed the variance of seed FC maps across individuals into principal components using the scikit-learn (*Abraham et al., 2014*) PCA implementation. We computed the ratio of variance explained by each principal component and represented each principal component in volumetric voxel space for visual inspection across the seed networks.

## Replicability

We tested the replicability of the associations between seed FC FC subtypes and ASD diagnosis in an independent replication sample. Within the replication sample we computed individual seed FC maps for the 18 non-cerebellar MIST_20 seed networks, centered the seed FC maps to the replication sample group average and regressed variance of non-interest due to age, head motion and imaging site for each voxel. For the residual seed FC maps, we computed the continuous FC subtype assignment scores with the FC subtypes identified in the discovery sample. For those FC subtypes that showed significant associations with ASD diagnosis in the discovery sample, we then investigated the difference in continuous FC subtype assignment scores between ASD and NTC individuals in the replication sample.

## Acknowledgements

This research was supported by computation resources of Calcul Quebec and Compute Canada. We thank Gleb Bezgin, Budhachandra Khundrakpam, Yasser Iturria Medina, and John Lewis for helpful discussions of the analytic concept. For their feedback on the writing of this manuscript we want to thank Julie Boyle, Nida Ali, and Jonas Nitschke. We thank the ABIDE consortium and the consortium for reliability and reproducibility (CORR) for making publicly available the large datasets that this study was based on.

# Additional information

## Funding

| Funder | Grant reference number | Author |
| --- | --- | --- |
| Australian Research Council | DE170101134 | Hien Duy Nguyen |
| Brain Canada Multi Investigator Research Initiative (MIRI) and Azrieli Foundation | 3388 | Sebastian GW Urchs |
| Canadian Open Neuroscience Plastudenttform | Student Scholar Award | Sebastian GW Urchs |
| Courtouis Neuromod Foundation | Graduate Student Funding | Sebastian GW Urchs Angela Tam Clara Moreau Yassine Benhajali |
| Fonds de Recherche du Québec - Santé | Salary Award | Pierre Orban |
| Centre de recherche de l'Institut universitaire de geriatrie de Montreal | Graduate Student Funding | Angela Tam Yassine Benhajali |
| Canadian Consortium on Neurodegeneration in Aging | Graduate Student Funding | Clara Moreau Yassine Benhajali |
| Healthy Brains, Healthy Lives | Graduate Student Funding | Clara Moreau |
| Brain Canada Multi investigator research initiative | Graduate Student Funding | Clara Moreau |
| Australian Research Council | DP180101192 | Hien Duy Nguyen |

The funders had no role in study design, data collection and interpretation, or the decision to submit the work for publication.

## Author contributions

Sebastian GW Urchs, Conceptualization, Data curation, Software, Formal analysis, Validation, Investigation, Visualization, Methodology, Writing – original draft, Writing – review and editing; Angela Tam, Conceptualization, Software, Methodology, Writing – review and editing; Pierre Orban, Conceptualization, Methodology, Writing – review and editing; Clara Moreau, Writing – review and editing; Yassine Benhajali, Data curation, Validation, Writing – review and editing; Hien Duy Nguyen, Methodology, Writing – review and editing; Alan C Evans, Conceptualization, Resources, Supervision, Funding acquisition, Project administration, Writing – review and editing; Pierre Bellec, Conceptualization, Resources, Software, Supervision, Funding acquisition, Methodology, Project administration, Writing – review and editing

## Author ORCIDs
Sebastian GW Urchs ⓘ http://orcid.org/0000-0001-5504-8579
Hien Duy Nguyen ⓘ http://orcid.org/0000-0002-9958-432X

## Ethics

Human subjects: All human imaging data used in this study were sampled from publicly available datasets. The inclusion of data in these samples were conditional on the approval of the respective local Institutional Review Board (IRB) and were shared in a de-identified form according to the requirements identified by the Health Insurance Portability and Accountability Act (HIPAA). The use of these data for the analyses presented in this study were approved by the ComitéMixte d'éthique en recherche regroupement neuroimagerie du Québec (CMER RNQ) approval number 14-15-002.

Decision letter and Author response
Decision letter https://doi.org/10.7554/eLife.56257.sa1
Author response https://doi.org/10.7554/eLife.56257.sa2

## Additional files

### Supplementary files
• Transparent reporting form

### Data availability
All imaging data used in this study are publicly available from their respective dataset repositories: http://fcon_1000.projects.nitrc.org/indi/abide/ for ABIDE 1 and 2; http://fcon_1000.projects.nitrc.org/indi/CoRR/html/hnu_1.html for HNU 1. Source data have been provided for figure 2. The analysis code is available at https://github.com/surchs/ASD_subtype_code_supplement, (copy archived at swh:1:rev:8ed67ada3cdceb2a45b53e5d69b2a0a8cd6035f1).

The following previously published datasets were used:

| Author(s) | Year | Dataset title | Dataset URL | Database and Identifier |
|---|---|---|---|---|
| Di Martino A | 2014 | The Autism Brain Imaging Data Exchange: Towards a Large-Scale Evaluation of the Intrinsic Brain Architecture in Autism | http://fcon_1000.projects.nitrc.org/indi/abide/abide_I.html | 1000.projects, abide_I |
| Di Martino A | 2017 | Enhancing Studies of the Connectome in Autism Using the Autism Brain Imaging Data Exchange II | http://fcon_1000.projects.nitrc.org/indi/abide/abide_II.html | 1000.projects, abide_II |
| Weng X, Zuo X, Chen B, Ge Q | 2014 | Hangzhou Normal University Dataset 1 | https://doi.org/10.15387/fcp_indi.corr.hnu1 | NITRC, 10.15387/fcp_indi.corr.hnu1 |

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

## Appendix 1

### Subtype Topography

Topographic overview of all FC-subtypes extracted from the discovery sample. Each row corresponds to the 18 non-cerebellar MIST_20 seed networks. Left column: a glass brain representation of the seed network (in green), and the average seed FC map in the discovery sample before nuisance covariate regression. Right column: extracted FC-subtype maps in arbitrary order. The green outline within each subtype map represents the outline of the seed network. Numbers below each subtype map reflect from left to right: the total number of individuals with discrete assignment to the subtype in the discovery sample, the total number of ASD individuals with discrete assignments to the subtype, the percentage of ASD individuals assigned to this subtype. For all subtypes, the chance level of assignment to the subtype is 50% because the discovery sample has the same number of ASD and NTC individuals. Seed networks have been grouped according to their functional hierarchy in the MIST atlas.

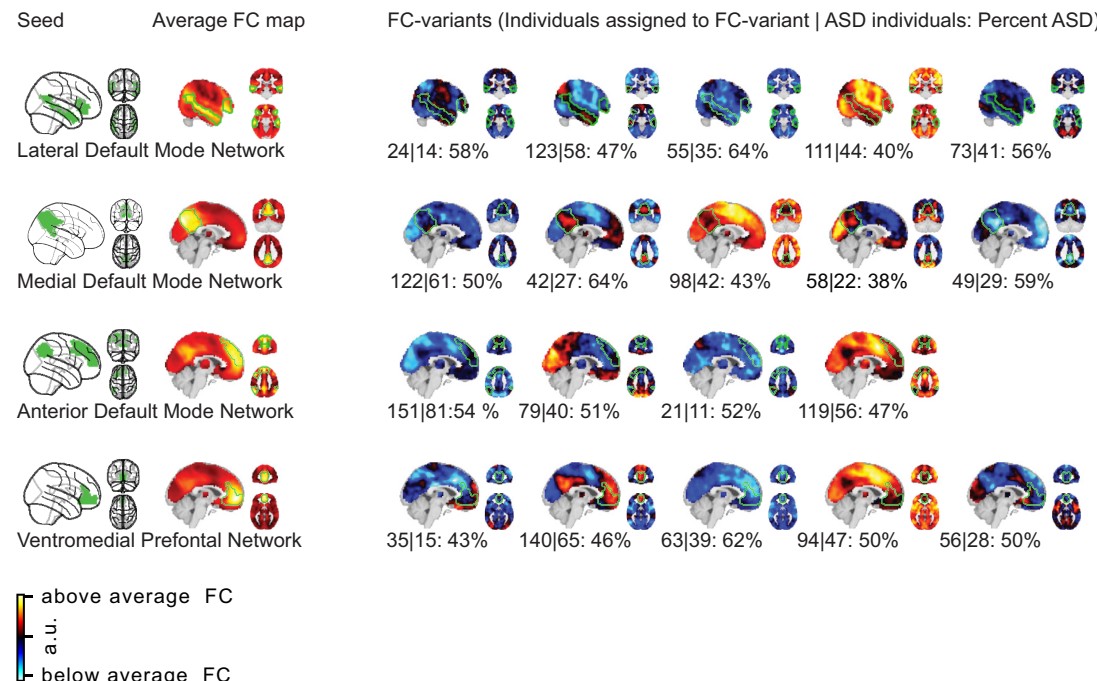

**Appendix 1—figure 1.** Topography of FC subtypes for Default Mode Network seed networks.

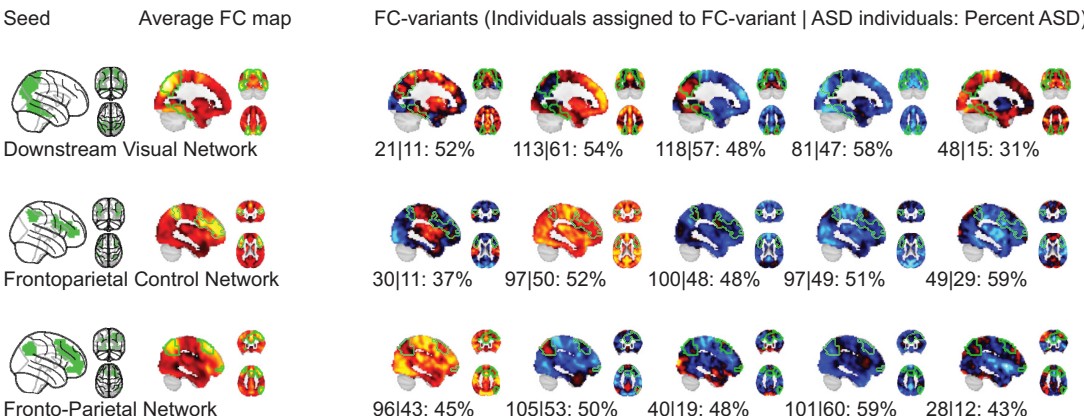

**Appendix 1—figure 2.** Topography of FC subtypes for seed networks in the frontoparietal control and downstream visual networks.

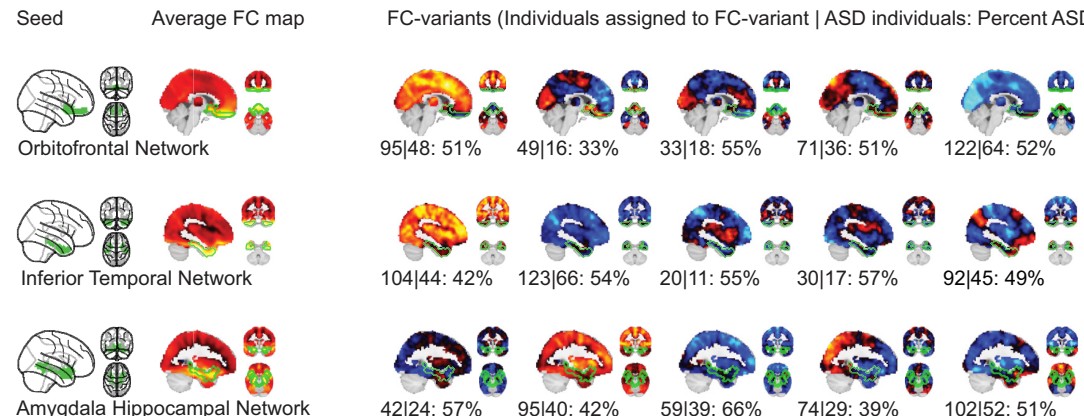

**Appendix 1—figure 3.** Topography of FC subtypes for seed networks in subcortical, orbitofrontal, and temporal networks.

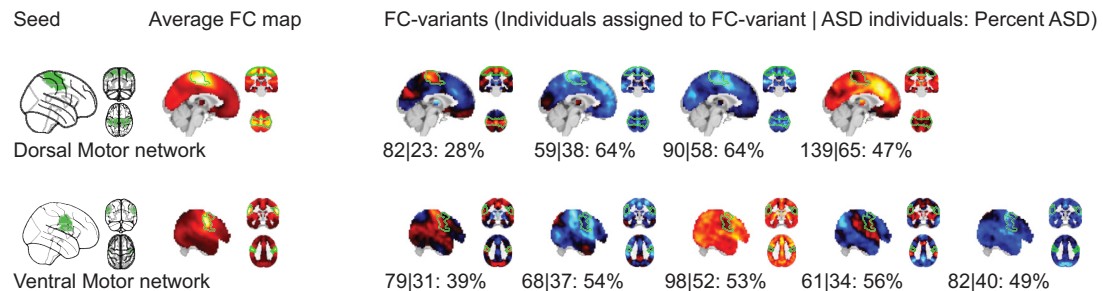

**Appendix 1—figure 4.** Topography of FC subtypes for motor seed networks.

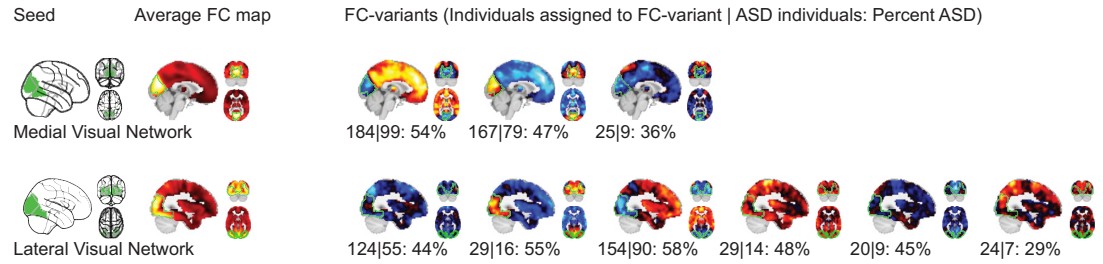

**Appendix 1—figure 5.** Topography of FC subtypes for visual seed networks.

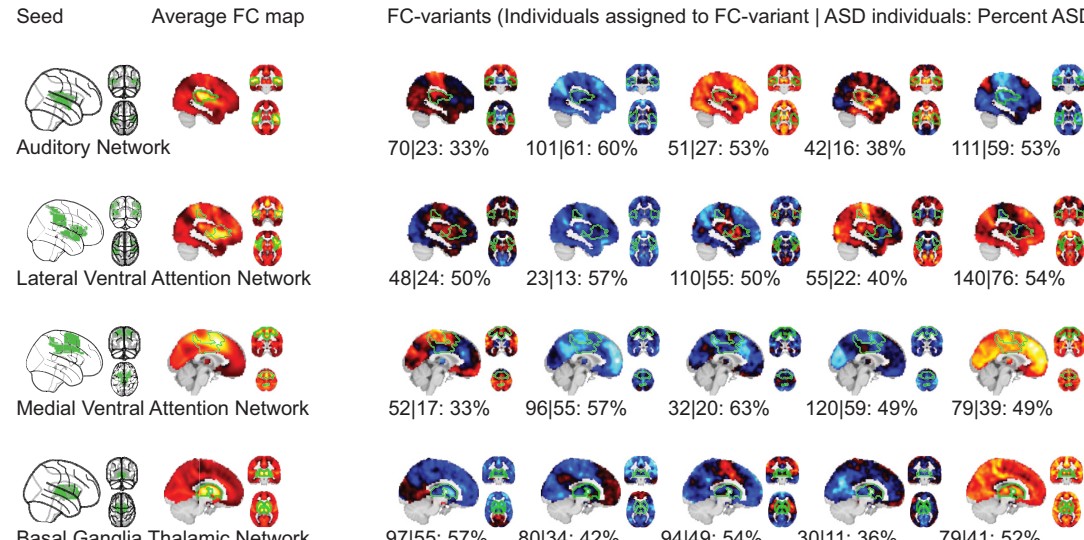

Seed Average FC map FC-variants (Individuals assigned to FC-variant | ASD individuals: Percent ASD)

Auditory Network 70|23: 33% 101|61: 60% 51|27: 53% 42|16: 38% 111|59: 53%

Lateral Ventral Attention Network 48|24: 50% 23|13: 57% 110|55: 50% 55|22: 40% 140|76: 54%

Medial Ventral Attention Network 52|17: 33% 96|55: 57% 32|20: 63% 120|59: 49% 79|39: 49%

Basal Ganglia Thalamic Network 97|55: 57% 80|34: 42% 94|49: 54% 30|11: 36% 79|41: 52%

**Appendix 1—figure 6.** Topography of FC subtypes for seed networks in the ventral attention and salience networks.

## Appendix 2

### Robustness of ASD subtypes

Overview of the robustness of subtype associations with ASD diagnosis to changes in the distance cutoff parameter. (A) Left: Average percentage of sample assigned to any subtype (black line) and average number of identified subtypes (orange line) for different levels of distance cutoff parameters. Shaded areas show range of values across all seed networks. Note the sharp drop of both metrics for more stringent distance cutoff parameters (black to light grey shaded values on the horizontal axis). Right: Correlation of ASD effect sizes in discovery and replication sample across different remains largely unaffected by cutoff values. (B) Average subtype maps for negASD (top row) and posASD (bottom row) subtypes, spatial patterns are highly preserved across thresholds. (C) Breakdown of individual subtype maps across different levels of thresholds illustrated by the example of the dorsal somato-motor seed network. (D) Breakdown of subtypes across threshold levels illustrated by the example of the subject by subject dissimilarity matrix of the dorsal somato-motor seed network. Grey shaded overlays reflect the subtype solutions at different dissimiliarity thresholds. MIST_20 seed network names are abbreviated, see *Table 1* for full seed network names.

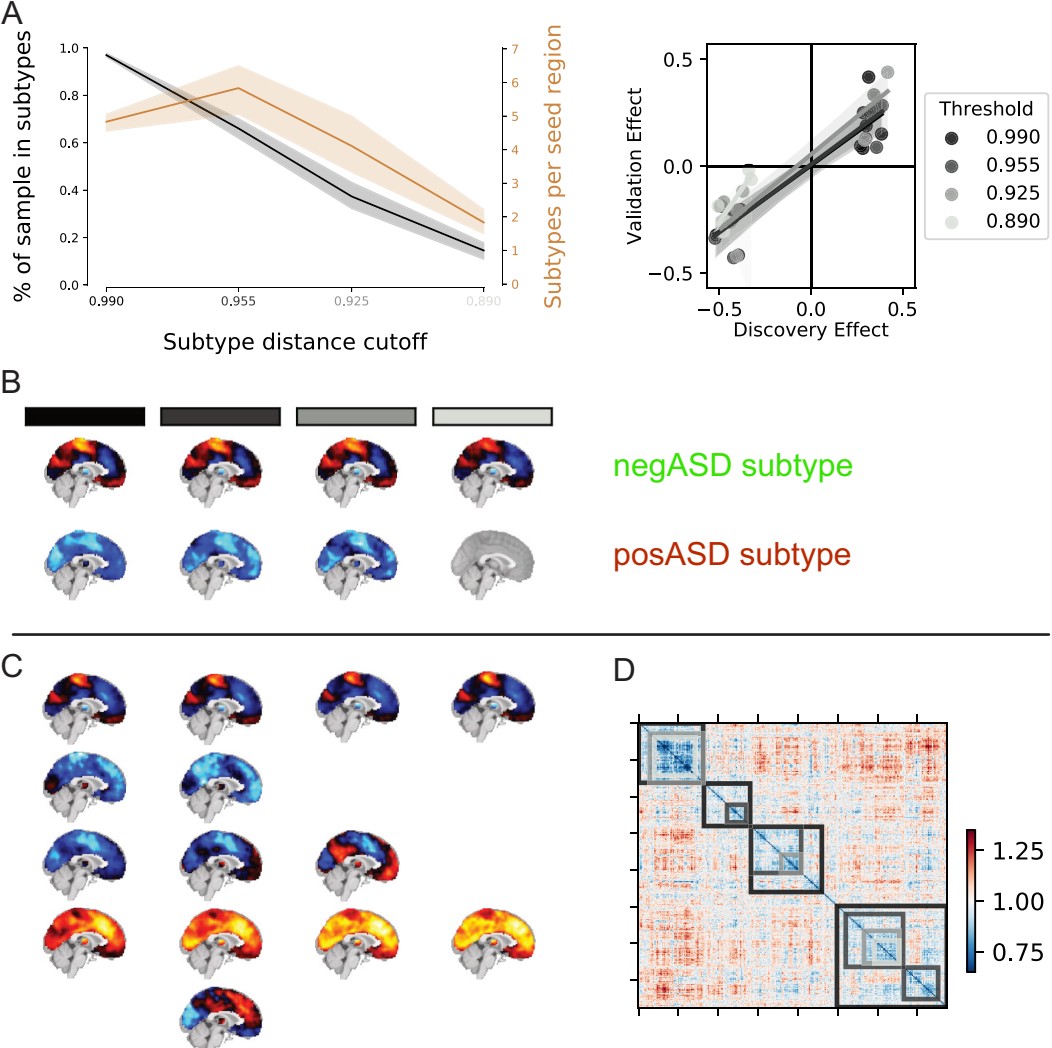

**Appendix 2—figure 1.** Subtype Robustness.

# Appendix 3

## PCA Components

Spatial maps of subtypes with significant association to ASD diagnosis for supplementary analysis that regressed global connectivity from all seed FC maps. Subtypes with significant negative association of continuous assignments and ASD diagnosis are shown on the left, those with positive associations on the right. PCA components have been grouped according to the functional hierarchy of their corresponding seed networks in the MIST atlas.

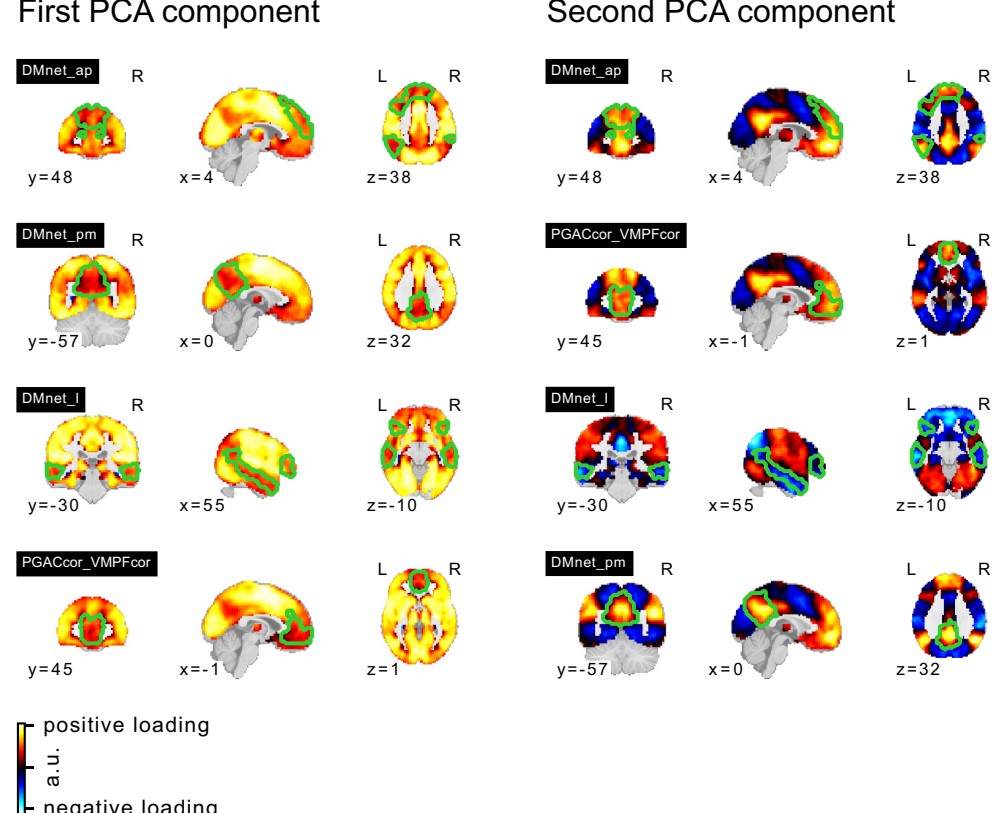

**Appendix 3—figure 1.** PCA components of Default Mode Network seed networks.

## First PCA component Second PCA component

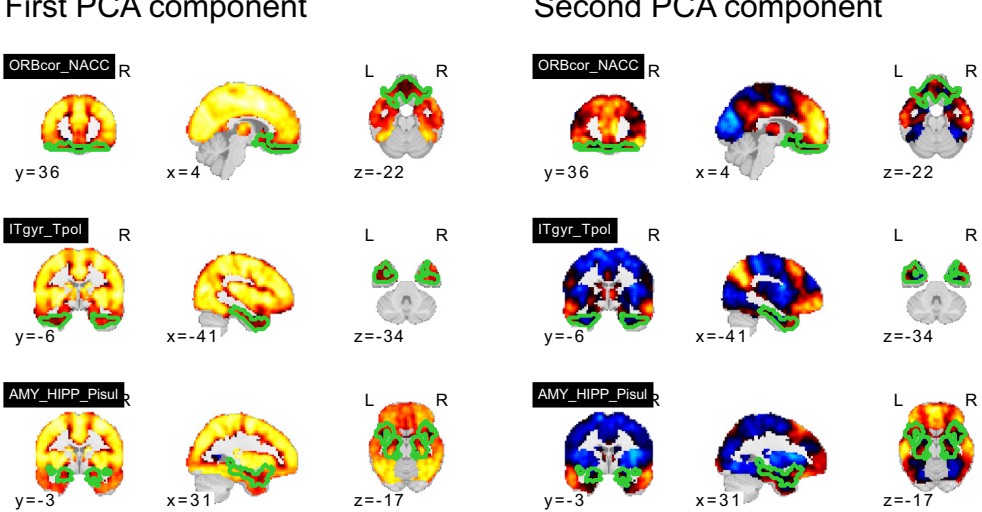

**Appendix 3—figure 2.** PCA components of seed networks in the frontoparietal control and downstream visual networks.

## First PCA component Second PCA component

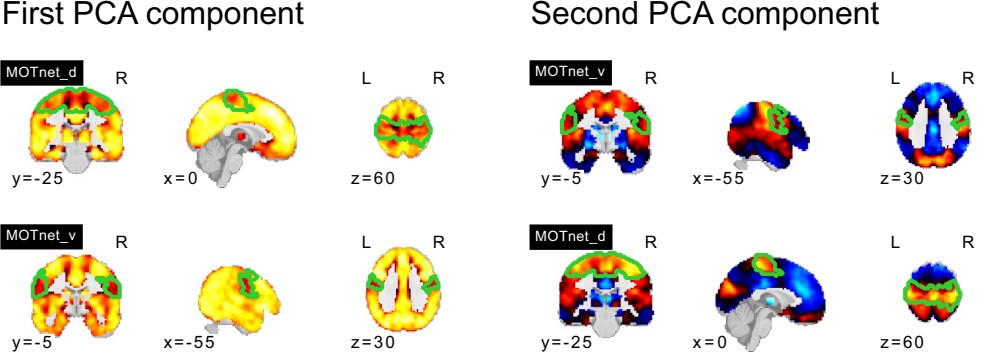

**Appendix 3—figure 3.** PCA components of seed networks in the subcortical, orbitofrontal, and temporal networks.

## First PCA component Second PCA component

**Appendix 3—figure 4.** PCA components of motor seed networks.

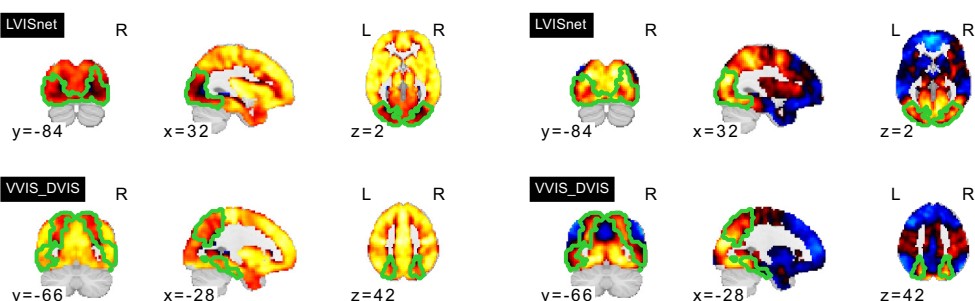

**Appendix 3—figure 5.** PCA components of visual seed networks.

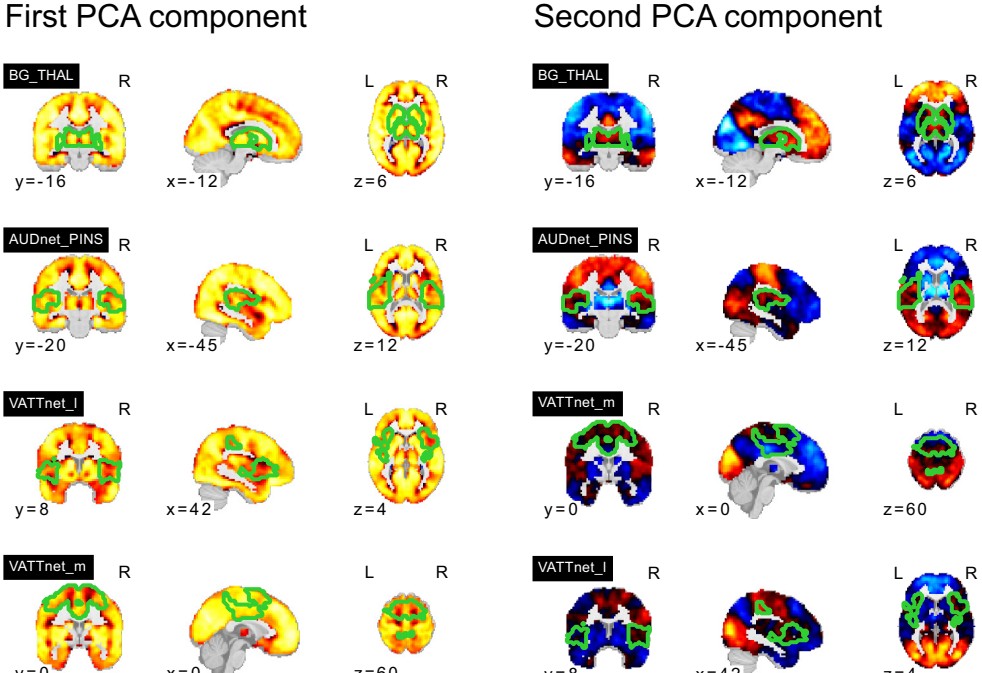

**Appendix 3—figure 6.** PCA components of seed networks in the ventral attention and salience networks.

