## [Editor Report]

The authors examine autism subtypes using functional connectivity data derived from magnetic resonance imaging. Autism spectrum disorder is notoriously heterogeneous, so the clustering approach to decompose this heterogeneity is attractive, however, the robustness of this approach and the generalization of groupings is unknown. The authors find that functional connectivity subtypes correspond to clinical autism diagnostic groupings and generalize using independent replication data. Functional connectivity patterns are robust, but the discrete assignment of individuals to a group is moderate and suggests that the findings may reflect compression of the primary gradient of functional brain organization.

---

## [Decision Letter]

**Decision letter after peer review:**

Thank you for submitting your article "Subtypes of functional connectivity associate robustly with ASD diagnosis" for consideration by *eLife*. Your article has been reviewed by 3 peer reviewers, and the evaluation has been overseen by a Reviewing Editor and Christian Büchel as the Senior Editor. The following individual involved in review of your submission has agreed to reveal their identity: Ralph-Axel Müller (Reviewer #3).

The reviewers have discussed the reviews with one another and the Reviewing Editor has drafted this decision to help you prepare a revised submission.

Summary:

The study examines how to subtype individuals with/without an ASD diagnosis using functional connectivity (FC) data. ASD is notoriously heterogenous. As such, a clustering approach to decompose this heterogeneity is attractive but the robustness of this approach and the generalizability of the identified groups is unknown. Urchs et al., find that FC subtypes correspond to ASD diagnosis and generalize using independent replication data. They show that FC patterns are robust, but discrete assignment of individuals to a group are moderate. The study finds that FC patterns for some ROIs are moderately linked with ASD diagnostic status. Particular strengths of the study lie in the great length it goes to achieve replication and reliability (using multiple separate datasets), and to protect from confounds. Reviewers also appreciated the clear conceptual distinction in this study between discrete and continuous heterogeneity within the ASD population, which is often not explicitly addressed in studies testing for "subtypes" or "clusters". Other interesting aspects of the study are the support for 'dense sampling' (i.e., low reliability of datasets with only a single short fMRI scan per participant). Overall, the manuscript presents a relevant and well thought out piece of work. However, as written the analyses are difficult to follow, there was confusion from Reviewers as to the terminology and biological relevance of 'subtypes'. As currently presented, the paper emphasizes evaluation of the reproducibility of the subtypes identified, but the neurobiological validity, clinical utility and potential new leads into functional brain architecture in ASD are unclear. Please see specific comments below.

Essential revisions:

1. It may be possible that the dissimilarity matrices are not optimally modelled in terms of discrete clusters. Testing should be undertaken to determine whether discrete clusters are indeed present in the data, to ensure that the clustering algorithm is not simply forcing clusters onto variation that is truly continuous. Given that the continuous assignment provided better replicability, it is possible that a continuous representation of the subtypes also provides a better characterization.

2. The protective and risk subtypes appear to parse out topographic patterns of global above- and below-average functional connectivity respectively (Figure 3). This would suggest that the subtypes are tapping into a basic FC property, such as the global fMRI signal or whole-brain averaged connectivity. Authors should characterize the topography of the subtypes, to rule out global effects. The major concern in this regard is that the topography of the subtypes are not discussed in any detail, and the abbreviations in Figure 3 are not defined. The methodological approach seems a bit peculiar. FC maps were created for 18 seeds, but justification is not sufficient. Selection of ROIs and how networks likely impact patterns of findings, so the rationale for this choice, as opposed to more common parcellation schemes (e.g., Power, Gordon, Glasser, Schaefer), needs better justification.

3. The "FC subtypes" were determined for each of the 18 seeds separately, resulting in a plethora of "subtypes". The authors later note that the subtypes for some seeds show similarities. Of course, this would be expected because they may actually reflect the same network patterns, being captured via different seeds from the catalogue of 18. Seed selection adds a potentially arbitrary step early in the pipeline that may weaken the study in comparison to data-driven approaches (e.g., ICA).

Generally, the use of the term "subtype" may be misleading, as readers may expect more evidence of ASD "subtypes". The paper has at least one important message about these: They are probably not discrete, but there is continuity between different variants of the disorder, as far as fcMRI patterns go. But the "subtypes" described in the paper are not subtypes of the disorder, and may only be rather indirectly related to them.

4. There is a big methodological step from the detection of 87 "FC subtypes" to the focus on only 11 that show significant association with ASD. Significance thresholds are ultimately arbitrary statistical lines in the sand that may not adequately reflect underlying biology. Readers will want to know more about this vast majority of 76 "subtypes" that are discarded from further analysis. It is extremely likely that these have some informative value for diagnostic classification, even if each of their singular "FC subtypes" does not 'significantly' link up with diagnostic status.

5. Could the no-added effect of symptom severity (on top of diagnosis) be attributable to there being a much greater effect from ASD vs. NTC? What do these results look like if you only include ASD individuals in your analysis? Are 'different' communities identified that do have an association with symptom severity in this approach? Essentially, are your community assignments dominated by ASD vs. control and can you find more subtle ASD symptom relevant communities when you leave out NTC individuals?

6. Authors choose ADOS which is limited to ASD individuals – why not look at SRS which is available from many NTC individuals? Wouldn't this behavior metric be more relevant to the present analysis since this study focused on NTC vs. ASD? It would be interesting to look at association with ADOS repeating the analyses with ASD individuals only, and to look at SRS scores when NTC and ASD individuals are included.

7. When comparing across multiple sessions, are the individuals 'in question' left out of the community group mean? Individuals are identifiable by their connectivity data. Thus, is having an individual's scan included in the community mean biasing the result of that individual being most similar to the 'right' community?

8. It is likely, as authors point out, that more data is driving their results (Figure 1 right). However, more sessions, states sampled, could also be contributing. This can be explicitly tested by holding the total amount of data constant and looking at improvement collapsing across more sessions. The authors are in a good position to comment on this debate using the data they have.

9. The stunning correlation depicted in Figure 2d is misleading. A bimodal distribution of datapoints is enforced as only 11 of 87 FC subtypes in the two tails (significant association with diagnostic status, either ASD or TD) are included. The high Pearson correlation for discovery vs. replication datasets is an artifact of that. Test for all 87 subtypes, or for TD vs. ASD associated subtypes separately, and the correlation will be more realistic. No big deal anyway because authors admit the diagnostic association and its replication are moderate. But authors should remove or edit the corresponding passage in the Discussion (l.292)

---

## [Author Response]

Essential revisions:1. It may be possible that the dissimilarity matrices are not optimally modelled in terms of discrete clusters. Testing should be undertaken to determine whether discrete clusters are indeed present in the data, to ensure that the clustering algorithm is not simply forcing clusters onto variation that is truly continuous. Given that the continuous assignment provided better replicability, it is possible that a continuous representation of the subtypes also provides a better characterization.

Thank you for the suggestion, we implemented the proposed experiment by conducting a principal component analysis. Briefly, the first component appeared to capture global changes in FC, while the second seemed to recap the extrinsic vs intrinsic network decomposition, originally described in (Goland et al., 2008) and recently rediscovered as “gradient” (Margulies et al., 2016), and further shown to be associated with ASD (Hong et al. 2019). This second component, presented below and further called PCA gradient, is convergent with several of our FC-FC subtype maps, so both analytical techniques (clustering vs PCA) seem to capture consistent dimensions of variations, at least for the main components. The remaining principal components only appeared to explain marginal amounts of variance and were not investigated further. We have added a section explaining this supplementary analysis to the Methods (l. 795-801), discussion (l. 474-490) sections and Annex 3 (l. 1064-1092). We are also mentioning this important observation in our abstract (l. 27-29) and conclusions (l. 533-535)

2. The protective and risk subtypes appear to parse out topographic patterns of global above- and below-average functional connectivity respectively (Figure 3). This would suggest that the subtypes are tapping into a basic FC property, such as the global fMRI signal or whole-brain averaged connectivity. Authors should characterize the topography of the subtypes, to rule out global effects. The major concern in this regard is that the topography of the subtypes are not discussed in any detail, and the abbreviations in Figure 3 are not defined. The methodological approach seems a bit peculiar.

Regarding the discussion of the topography of the FC subtypes, we have added a series of figures in annex 1 (Figures 1 to 6) recapitulating all the 87 FC subtypes for the different seeds.

Regarding the influence of global FC variations, we repeated the subtyping procedure, now called FC subtype analysis, after regressing the average brain connectivity across subjects, at each voxel independently, following the “global mean regression” approach described in (Yan et al., 2013, https://doi.org/10.1016%2Fj.neuroimage.2013.04.081).

The negASD FC subtypes are left mostly unchanged, and resemble the “gradient” from PCA. But the posASD FC subtypes become more clearly a mirror of the negASD FC subtypes, therefore also converging on the “gradient” identified by the PCA. So it looks like, in the absence of “global mean regression”, FC subtypes capture additive effects of both global changes and the PCA gradient.

We added a paragraph on these analyses in the Results (l. 324-341, Figure 6), Methods (l. 646-649) and Discussion (l. 471-475) sections.

FC maps were created for 18 seeds, but justification is not sufficient. Selection of ROIs and how networks likely impact patterns of findings, so the rationale for this choice, as opposed to more common parcellation schemes (e.g., Power, Gordon, Glasser, Schaefer), needs better justification.

Our rationale was the availability of carefully multi-resolution labels, which none of the other atlases offer (including Schaefer). Regarding the choice of the scale, we selected the one available in MIST which was the lowest possible (as we examined every network separately), while separating the cortical and subcortical network components. We also noted that the MIST parcellation appears to perform on par or superior to other templates on a number of prediction benchmarks implemented independently from our group, and in particular regarding ASD prediction (Dadi et al., 2020). While we were revising this work, a new machine learning evaluation of rs-fMRI for ASD was published, which also found that MIST performs well on this task (https://www.medrxiv.org/content/10.1101/2021.10.21.21265162v1.full.pdf). We mention this justification in the introduction (l. 102-104).

3. The "FC subtypes" were determined for each of the 18 seeds separately, resulting in a plethora of "subtypes". The authors later note that the subtypes for some seeds show similarities. Of course, this would be expected because they may actually reflect the same network patterns, being captured via different seeds from the catalogue of 18. Seed selection adds a potentially arbitrary step early in the pipeline that may weaken the study in comparison to data-driven approaches (e.g., ICA).

We would like to clarify that the 18 seeds were selected based on the MIST atlas, which is itself generated through a fully data-driven procedure on an independent fMRI sample. So our approach is very similar to using an ICA template, except the parcels were generated via cluster analysis, and that we opted for seed-based correlation maps rather than dual regression, as is often done using ICA. By construction, these 18 seeds identify different network patterns because those differences are what drives the cluster analysis. We could have investigated decompositions with lesser granularity, but we were interested to see if association between FC subtypes and specific symptoms such as ADOS would emerge specifically in some networks, which turned out not to be the case. We added this hypothesis in the Abstract (l. 25-29), Introduction (l. 98-104), and noted in the results (l. 305-323) and discussion that the findings were not consistent with our hypothesis (l. 465-492).

Generally, the use of the term "subtype" may be misleading, as readers may expect more evidence of ASD "subtypes". The paper has at least one important message about these: They are probably not discrete, but there is continuity between different variants of the disorder, as far as fcMRI patterns go. But the "subtypes" described in the paper are not subtypes of the disorder, and may only be rather indirectly related to them.

We thank the reviewer for this comment. In response, the manuscript now consistently uses the term “FC subtypes” rather than just “subtypes” to make clear these subtypes are a pure reflection of functional connectivity and do not relate to clinical manifestations.

4. There is a big methodological step from the detection of 87 "FC subtypes" to the focus on only 11 that show significant association with ASD. Significance thresholds are ultimately arbitrary statistical lines in the sand that may not adequately reflect underlying biology. Readers will want to know more about this vast majority of 76 "subtypes" that are discarded from further analysis. It is extremely likely that these have some informative value for diagnostic classification, even if each of their singular "FC subtypes" does not 'significantly' link up with diagnostic status.

We agree that it is useful to visualise the topography of all subtypes, and not only those passing a statistical test of association with ASD. We added a new series of figures in Annex 1 (Figures 1-6) that illustrates all 87 FC subtype maps, together with the number and diagnosis of the contributing individuals. We also agree that the non-significant subtypes may carry relevant information. We added this point to the discussion (l. 487-492). We investigated this idea using an ensemble predictive technique in a separate study, available as a preprint (Urchs *et al.*, 2020).

5. Could the no-added effect of symptom severity (on top of diagnosis) be attributable to there being a much greater effect from ASD vs. NTC? What do these results look like if you only include ASD individuals in your analysis? Are 'different' communities identified that do have an association with symptom severity in this approach? Essentially, are your community assignments dominated by ASD vs. control and can you find more subtle ASD symptom relevant communities when you leave out NTC individuals?

Thanks for the suggestion. We replicated the main experiments by subtyping only the participants on ASD, and could not identify subtypes which were associated with symptoms severity as measured by the ADOS. This additional experiment was added to the manuscript (l. 273-304).

6. Authors choose ADOS which is limited to ASD individuals – why not look at SRS which is available from many NTC individuals? Wouldn't this behavior metric be more relevant to the present analysis since this study focused on NTC vs. ASD? It would be interesting to look at association with ADOS repeating the analyses with ASD individuals only, and to look at SRS scores when NTC and ASD individuals are included.

We also tested for the association between total SRS scores and continuous assignments to FC subtypes identified in the discovery sample. SRS scores are available for some NTC individuals in our sample (N total = 199, N ASD = 108). We did find one FC subtype based on the ventro-medial prefrontal seed region that showed a significant association with raw SRS scores. Further description of this result can be found in the Results section (l. 273-304).

7. When comparing across multiple sessions, are the individuals 'in question' left out of the community group mean? Individuals are identifiable by their connectivity data. Thus, is having an individual's scan included in the community mean biasing the result of that individual being most similar to the 'right' community?

This reviewer is correct; a subject’s data is used to derive the average of its subtype, although this could possibly induce a bias, although likely small for subtypes featuring large numbers of subjects. However, to compute ICC coefficients, we need to derive all measures using the same FC subtype. We acknowledged this limitation in the Discussion section (l. 514-518).

8. It is likely, as authors point out, that more data is driving their results (Figure 1 right). However, more sessions, states sampled, could also be contributing. This can be explicitly tested by holding the total amount of data constant and looking at improvement collapsing across more sessions. The authors are in a good position to comment on this debate using the data they have.

Thank you for the very helpful suggestion. We have conducted the proposed analysis by both manipulating the number of sessions and the number of included time points. In brief we found that mixing data from different sessions, while maintaining the total amount of time points constant, does increase mildly the reliability of subtypes. But the most massive gains come from using more time points. See Results section l. 196-204, Discussion l. 400-407 and Figure 3.

9. The stunning correlation depicted in Figure 2d is misleading. A bimodal distribution of datapoints is enforced as only 11 of 87 FC subtypes in the two tails (significant association with diagnostic status, either ASD or TD) are included. The high Pearson correlation for discovery vs. replication datasets is an artifact of that. Test for all 87 subtypes, or for TD vs. ASD associated subtypes separately, and the correlation will be more realistic. No big deal anyway because authors admit the diagnostic association and its replication are moderate. But authors should remove or edit the corresponding passage in the Discussion (l.292)

Deriving the correlation using all 87 FC subtypes indeed reduces the correlation between discovery and replication effect sizes, from 0.91 (using only the significant FC subtypes), down to 0.71 (using all 87 FC subtypes). We now include both analyses in the Results section (l. 353-355) and have updated the Discussion (l. 447-464) and Figure 7 panel d.